# VideoMathQA: Benchmarking Mathematical Reasoning via Multimodal Understanding in Videos

**Hanoona Rasheed**[1], **Abdelrahman Shaker**[1], **Anqi Tang**[1], **Muhammad Maaz**[1]
**Ming-Hsuan Yang**[2,3], **Salman Khan**[1,4], **Fahad Shahbaz Khan**[1,5]
[1]MBZUAI    [2]University of California Merced    [3]Google Research
[4]Australian National University    [5]Linköping University
https://mbzuai-oryx.github.io/VideoMathQA

## Abstract

Mathematical reasoning in real-world video presents a fundamentally different challenge than static images or text. It requires interpreting fine-grained visual information, accurately reading handwritten or digital text, and integrating spoken cues, often dispersed non-linearly over time. In such multimodal contexts, success hinges not just on perception, but on selectively identifying and integrating the right details from a rich and noisy stream of content. To this end, we introduce VideoMathQA, a benchmark designed to evaluate whether models can perform such temporally extended cross-modal reasoning on videos. The benchmark spans 10 diverse mathematical domains, covering videos from 10 seconds to over 1 hour. We employ graduate-level experts to ensure high quality, for over 920 man-hours of annotation. To reflect real-world scenarios, questions are designed around three core reasoning challenges: *direct problem solving*, *conceptual transfer*, which requires applying learned methods to new problems; and *deep instructional comprehension*, involving multi-step reasoning over extended explanations and partially worked-out solutions. Each question includes multi-step reasoning annotations, enabling fine-grained diagnosis of model capabilities. Through this benchmark, we establish an evaluation framework for models that must reason, rather than merely perceive, jointly ground concepts across visual, audio, and textual modalities, across temporally extended mathematical problem settings.

## 1 Introduction

Unlike mathematical reasoning benchmarks for static-images, videos present unique challenges. In educational and instructional videos, key information is conveyed through evolving diagrams, handwritten or digital notations, and spoken explanations that unfold non-linearly across time. A reasoning model must therefore sift through high-resolution frames, align visual representations with subtitles or voice-over, and integrate disparate cues into a coherent problem-solving pipeline. This *'needle-in-a-multimodal-haystack'* problem requires not only accurate perception (e.g., frame-aware OCR of equations) but also precision in symbolic manipulation and multi-step inference, where missing a single visual cue can lead to incorrect conclusions.

Existing benchmarks for mathematical reasoning (e.g., MathQA (Amini et al., 2019), ChartQA (Masry et al., 2022), and MathVista (Lu et al., 2024)) have driven substantial progress in image-based and text-based settings. However, they typically evaluate models on static diagrams, printed formulas, or single-turn queries and lack support for temporally extended contexts, dynamically evolving visual content, and do not offer true integration of vision, audio, text and background subject knowledge. Some recent efforts explore video question answering in general domains (Fu et al., 2025; Mangalam et al., 2024; Li et al., 2024d), but they do not target the precise, multi-modal, multi-step inference demanded by mathematical problem solving. Moreover, existing video benchmarks often rely on synthetic (Yi et al., 2019b) or narrowly scoped tasks (Shangguan et al., 2024) without detailed

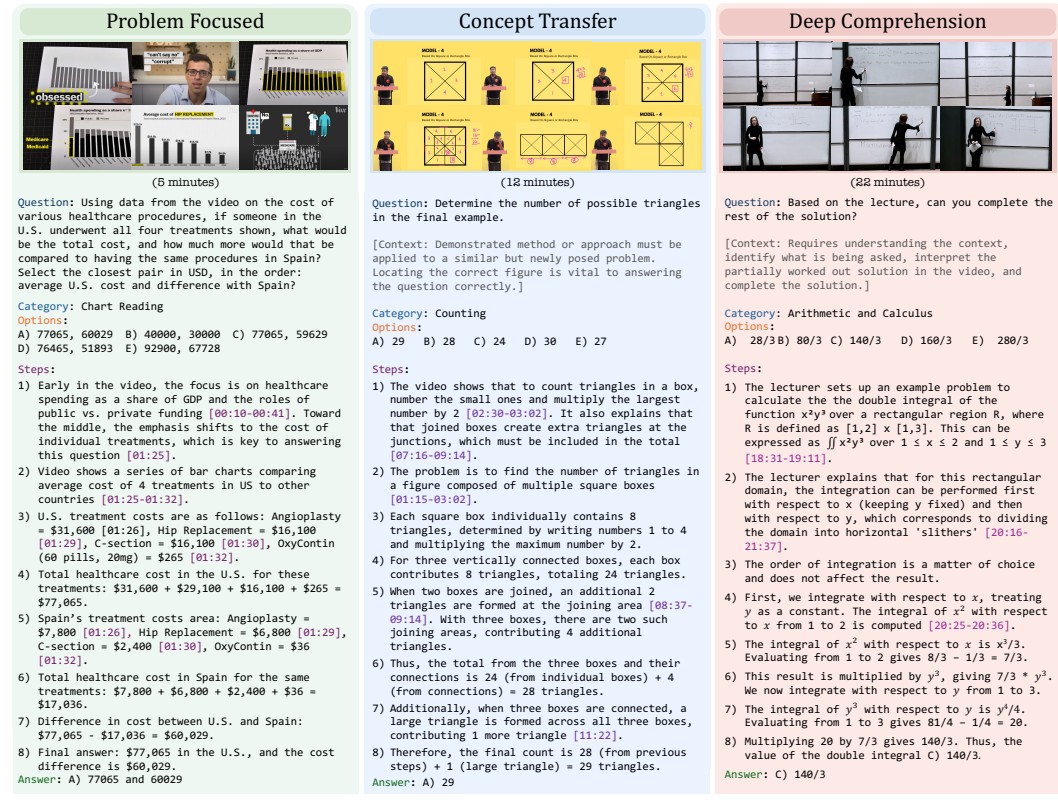

**Figure 1:** Examples from the VideoMathQA illustrating the three reasoning types: Problem Focused, Concept Transfer, and Deep Comprehension. Each sample includes a carefully selected video, question, five multiple-choice options, expert-annotated reasoning steps with temporal localization, and the final answer.

reasoning annotations, making it difficult to diagnose whether a model's solution stems from genuine logical inference or simple pattern matching.

To bridge these gaps, we introduce VideoMathQA, a comprehensive benchmark designed explicitly to evaluate deep mathematical reasoning in videos. It comprises manually annotated real-world video–question pairs drawn from various educational resources. VideoMathQA tests three core reasoning scenarios: *a) Direct Problem Solving*, where answers are grounded entirely in the presented content; *b) Conceptual Transfer*, requiring agents to apply learned methods to novel problems; and *c) Deep Instructional Comprehension*, involving multi-step reasoning over extended, partially worked-out explanations. Each instance includes high-resolution frames, aligned subtitles, and audio narration, and is annotated with fine-grained reasoning traces that capture every inference step to enable assessment of both intermediate steps and final outcomes. Videos span *ten mathematical domains* (e.g., geometry, calculus, statistics, graph theory, chart) and range from 10s clips to *hour-long* lectures, ensuring both short-term perception and long-range dependency evaluation (see Fig. 1 for illustrative examples of the reasoning scenarios). Our key contributions are:

- A multimodal video reasoning benchmark that demands precise integration of vision (high-resolution), text, audio and subject knowledge in solving complex math problems.

- Our benchmark offers three real-world task categories: problem solving, concept transfer, and deep comprehension, reflecting instructional scenarios in videos. It spans ten mathematical domains and environments (whiteboard scribbles, digital slides, animated charts), drawn from diverse sources.

- Fine-grained reasoning annotations and metrics for measuring both intermediate inference fidelity and final answer correctness, with explicit mechanisms to detect confabulations.

- A comprehensive evaluation across 30 models reveals that success relies not just on perception, but on sustained attention to subtle cues dispersed across time and modalities.

## 2 RELATED WORKS

The exploration of video understanding benchmarks has evolved rapidly alongside the developments of multimodal large language models (MLLMs). Early benchmarks such as ActivityNet-QA (Yu et al., 2019), NExT-QA (Xiao et al., 2021), and CLEVRER (Yi et al., 2019a) primarily supported models that preceded MLLMs, focused on short clips with short answers, targeting action recognition, causal reasoning, or temporal ordering. They offered a limited scope for open-ended reasoning or compositional language generation.

With the introduction of MLLMs, the need for more comprehensive evaluation protocols became apparent. VCGBench (Maaz et al., 2023) represented an early attempt to quantify MLLM performance by designing open-ended question answering benchmark built on ActivityNet videos. MVBench (Li et al., 2024d) proposed an extensive multiple-choice setup covering a wide range of perceptual and temporal reasoning tasks, though it was constrained to short video clips due to limitations in available datasets. Video-MME (Fu et al., 2025) addressed this gap by introducing a unified benchmark covering short, medium, and long videos. NExT-GQA (Xiao et al., 2024) expanded the evaluation space by adding grounded reasoning questions. TempCompass provided a controlled probe into fundamental temporal properties such as event order and speed. As models matured, research turned to the more demanding setting of long-form video understanding. EgoSchema (Mangalam et al., 2024) and MovieChat Song et al. (2024) emphasized narrative comprehension by extending question answering to minute-long (1-3 min) egocentric or movie clips, while LongVideoBench (Wu et al., 2024b), MLVU Zhou et al. (2025) and LVBench (Wang et al., 2024b) pushed the scale further to hour-long videos.

Beyond video duration, a separate line of work investigates benchmarks that emphasize knowledge-intensive and reasoning-centric challenges. WorldQA (Zhang et al., 2024c), Video-MMLU (Song et al., 2025), Video-MMMU (Hu et al., 2025c), and MMVU (Zhao et al., 2025) explicitly target high-level subject matter, from STEM lectures to professional training content, probing whether MLLMs can extract and apply domain expertise from video. VSI-Bench Yang et al. (2025b) complements this by diagnosing spatial intelligence, testing how well models infer layouts and object relations from egocentric perspectives.

While these benchmarks have advanced multimodal understanding in everyday contexts, they fall short of the deeper complexity posed by mathematical reasoning, where models must navigate non-linear content and demonstrate tightly integrated understanding across vision, audio, text, and background subject knowledge.

**Multimodal Mathematical Benchmarks**: Recent image-based mathematical benchmarks have significantly advanced the evaluation of multimodal models. MathVista (Lu et al., 2024) and Math-V (Wang et al., 2024a) include visual questions drawn from textbooks and competitions, while MMMU (Yue et al., 2024a) and MMMU-Pro (Yue et al., 2024b) introduce subject-specific questions with CoT prompts and OCR inputs. DynaMath (Zou et al., 2025) evaluates the robustness of mathematical reasoning through visual perturbations. Although these benchmarks span a diverse range of mathematical topics, they are fundamentally limited to assessing reasoning over static images, and the temporal dimension intrinsic to video, where mathematical information may unfold through lectures, stepwise derivations, or interactive explanations, is not captured.

| Benchmark | Domain | Dur. (s) | STEM | CoT | Annotation |
|---|---|---|---|---|---|
| CLEVRER Yi et al. (2019a) | Physics (Syn.) | 5 | ✓ | ✗ | Auto |
| MovieChat-1K Song et al. (2024) | Movie | 500 | ✗ | ✗ | Human |
| EgoSchema Mangalam et al. (2024) | Egocentric | 180 | ✗ | ✗ | Auto+Human |
| Video-MME Fu et al. (2025) | General | 1018 | ✗ | ✗ | Human |
| PerceptionTest Patraucean et al. (2023) | Perceptual | 23 | ✗ | ✗ | Human |
| MMBench-Video Fang et al. (2024) | General | ∼100 | ✗ | ✗ | Human |
| LongVideoBench Wu et al. (2024a) | General | 473 | ✗ | ✗ | Human |
| Video-MMMU Hu et al. (2025b) | Scientific | 506 | ✓ | ✗ | Human |
| **VideoMathQA(Ours)** | Math | 241 | ✓ | ✓ | **Human†** |

**Table 1:** Comparison of VideoMathQA with recent video-language benchmarks in terms of domain, duration, STEM focus, step-wise CoT annotation, and source. † Annotations for our VideoMathQA are provided by graduate-level human experts.

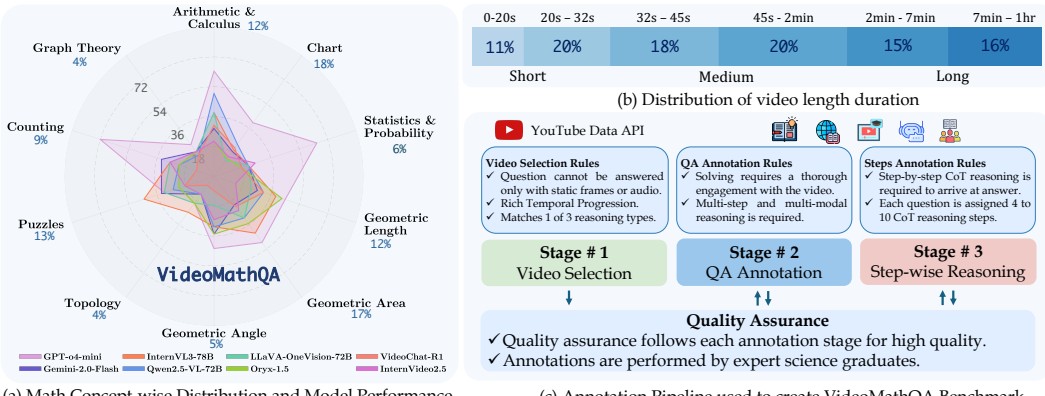

**Figure 2: a)** Question distribution and model accuracy across ten mathematical concepts in VideoMathQA, showing consistently low performance and highlighting current models' limitations. **b)** Video duration distribution, ranging from short 10s clips to long 1hr videos. **c)** Three-stage annotation in VideoMathQA, conducted by expert science graduates with detailed step-by-step reasoning and strict quality assessment.

Video-MMMU (Hu et al., 2025a) begins to explore video-based academic QA, but includes only a small subset of mathematical questions and focuses on general comprehension. It does not target the depth of reasoning or modality integration that mathematical problem solving demands. In contrast, VideoMathQA is designed to evaluate deep mathematical reasoning by challenging models to interpret high-resolution visuals, follow non-linear spoken explanations, and integrate vision, language, and domain knowledge over time, while providing detailed step-by-step reasoning annotations to enable comprehensive analysis beyond just final answer accuracy.

## 3 VIDEOMATHQA

Despite significant progress in image-based mathematical reasoning benchmarks, existing datasets fail to capture the unique challenges posed by video-based math problems. In designing VideoMathQA, we asked: *What core challenges must a benchmark capture to reflect real-world multimodal mathematical reasoning and understanding?* Our answers shaped the core principles of the benchmark. We identified *three critical challenges*: the need to interpret **dynamic visual representations**, such as diagrams constructed or modified over time, often involving handwritten or digital content requiring robust, frame-aware OCR; the need for **temporal reasoning over long, non-linear contexts**, where concepts unfold gradually or are revisited across time; and the need for **joint grounding across visuals, text, and audio**, as all three modalities often contribute distinct pieces of essential information. These interconnected challenges define the complexity of video-based mathematical tasks and directly inform the design of our benchmark.

VideoMathQA consists of 420 carefully curated video-question pairs drawn from diverse mathematical instructional content, including structured problem walkthroughs, concept demonstrations with follow-up questions, full-length whiteboard or digital lectures, and animated documentaries involving chart-based reasoning. Each question includes **multi-step reasoning steps**, with a total of **2,945 expert-annotated steps** across the dataset. These reasoning steps are annotated with *timestamps*, grounding major observations to their precise location in the video and enabling direct evaluation of whether models can align reasoning traces with temporal evidence. Each question is characterized along *four dimensions*: (*i*) **Mathematical concept**, covering 10 domains such as geometry, arithmetic and calculus, statistics and probability, counting, graph theory, puzzles, topology, and chart reading, (See Fig. 2a); (*ii*) **Reasoning type**, categorized as problem focused, concept transfer, or deep comprehension; (*iii*) **Video duration**, ranging from 10 seconds to over an hour and grouped as short, medium, or long, supporting evaluation of both short-term and long-range reasoning (See Fig. 2b); and (*iv*) **Difficulty level**, categorized as easy, medium, or hard (See Fig. 4b).

The annotation process consists of three stages: video selection, question-answer annotation, and step-wise reasoning (see Fig. 2c), requiring considerable expert effort. On average, 30 mins to identify a suitable video, 40 mins to craft a high-quality question-answer pair, and 1 hour to compose detailed step-by-step reasoning, resulting in approximately 2 to 2.5 hours per sample. Across the dataset,

this amounts to roughly 115 person-days of annotation by science graduates. Different annotators handled each stage of a sample to ensure independent verification, with each stage revising the quality of the previous one. This systematic revision process ensures the benchmark's overall quality and consistency.

## 3.1 VIDEO SELECTION

We curate video sources that capture core challenges, including dynamic visual representations, long-range temporal dependencies, and the need for joint grounding across visual, textual, and audio modalities. We do this by selecting videos where the associated questions cannot be reliably answered using static frames or audio transcripts alone, focusing on content with rich temporal progression such as step-by-step diagram construction and incremental equation derivation. Static slide presentations and videos with minimal visual transitions are excluded. All videos are sourced via the YouTube API, manually reviewed, and trimmed to retain only question-relevant segments. Selection is also guided by conceptual difficulty, ensuring coverage from high-school to advanced university-level and Olympiad-style problems.

For chart-based questions, we include animated charts from documentary and news sources, spanning bar plots (vertical and horizontal), pie charts, line graphs, histograms, and stacked bar charts. We prioritize videos with multiple charts whose interpretations are temporally and conceptually linked. For the remaining nine mathematical categories, we collect a diverse mix of lecture videos, screen recordings, and digital whiteboard tutorials that reflect instructional formats commonly used in educational content.

## 3.2 QUESTION-ANSWER ANNOTATION

Each video in VideoMathQAis paired with a carefully constructed multiple-choice question to evaluate a model's ability to reason over visual and temporal mathematical content. Questions are written ensuring that solving them requires meaningful engagement with the video, rather than relying on surface-level cues from transcripts or isolated frames. This stage also serves as quality assurance for video selection, with about 70% of videos retained, and the rest discarded. We categorize questions into three reasoning types: (i) *Problem Focused*, where the question is explicitly stated and can be solved through direct observation and reasoning; (ii) *Concept Transfer*, where a demonstrated method or solution approach must be applied to a similar but new problem; and (iii) *Deep Instructional Comprehension*, which involves following long instructional content, such as a lecture or tutorial, to understand the context, identify what is being asked, interpret the partially worked-out solution, and complete the solution. Fig. 1 illustrates an example of each reasoning type.

## 3.3 STEP-WISE REASONING AND QUALITY ASSESSMENT

Following question-answer annotation, each sample undergoes a second stage of annotation, where a separate annotator writes a *step-by-step explanation* detailing the reasoning to arrive at the final answer. Each question is assigned four to ten steps, each step reflecting a meaningful semantic progression, capturing a distinct, essential part of the overall solution. This results in a total of **2,945 high-quality expert-annotated reasoning steps**. Model-generated chain-of-thought responses are compared against these steps, enabling step-level evaluation of how far a model progresses toward the correct answer. This stage also serves as quality assurance for the question-answer pairs, allowing the annotator to verify them, correct any errors or improve clarity. Approximately 30% of the questions were refined during this stage. Additionally, each question is tagged with a set of likely error categories to support structured error analysis during the step evaluation. Finally, we conduct a review of the step annotations, during which 788 out of the 2,945 steps were revised.

## 3.4 COMPARISON WITH EXISTING VIDEO BENCHMARKS

While recent video-language benchmarks offer broad coverage across domains, they primarily focus on perceptual or narrative understanding, often lacking the depth, structure, and multi-modal precision required for mathematical reasoning. In contrast, VideoMathQA targets the *'needle-in-a-multimodal-haystack'* challenge by demanding tightly aligned interpretation across high-resolution visuals, spoken explanations, and textual content, characteristics underrepresented in prior works. Unlike benchmarks

that emphasize short-form clips or general comprehension, our dataset centers on domain-specific, step-intensive reasoning with annotated chain-of-thought traces, enabling fine-grained evaluation of both intermediate and final responses. We present a comparison of VideoMathQA with recent benchmarks in terms of domain, duration, STEM focus, and step-wise CoT annotation in the Appendix Tab. 1. By focusing on mathematical problem-solving in long instructional videos, it reveals challenges still underexplored in current video-language benchmarks.

## 4 EXPERIMENTS

We evaluate a diverse set of models on VideoMathQA, including **5 *proprietary multimodal models*** and **25 *open-source models*** selected for their strong capabilities in video reasoning. We specifically focus on state-of-the-art video models, spanning **4 *model size categories***: 5B, 9B, 40B, and 80B parameters. We cover Qwen2.5-VL (Bai et al., 2025), InternVL2.5 (Chen et al., 2024), InternVL3 (Zhu et al., 2025), PLM-LLaMA (Cho et al., 2025), Oryx-1.5 (Liu et al., 2024), LLaVA-OV (Li et al., 2024b), LongVA-DPO (Zhang et al., 2024a), InternVideo2.5 (Wang et al., 2025), and Aria (Li et al., 2024c). In addition, we include recent ***reasoning models*** that supporting chain-of-thought prompting (VideoChat-R1 (Li et al., 2025), Video-R1 (Feng et al., 2025)). To contextualize the performance of video-specialized models, we include **2 *baselines***, one vision-blind LLM (Qwen-2.5), and one state-of-the-art image model (Qwen2.5-VL) evaluated on a single key frame. We also include a human evaluation to ground the model results against a meaningful reference.

### 4.1 EVALUATION STRATEGIES

**Inference Protocol:** Since the benchmark emphasizes the importance of multimodal grounding and temporal reasoning across modalities, we ensure fairness by adapting input configurations to align with the strengths and operational capabilities of each model. Specifically, each model samples frames according to its optimal setting (32 for LLaVA-OneVision, up to 768 for Qwen2.5-VL, and full-video access for Gemini). Consequently, subtitles are aligned with the sampled frames, providing richer audio context to models capable of handling longer temporal sequences. Our setup rewards models that can leverage extended temporal context and effectively integrate multimodal information. Please refer to Appendix B for additional implementation details and Appendix E for the prompts used in CoT, postprocessing, step-wise evaluation and error analysis.

We implement ***four evaluation strategies*** to comprehensively assess model performance: (***i***) **Multiple-Choice Evaluation (MCQ)**: Each question provides five options (one correct and four distractors). This direct evaluation format offers clear reproducibility without reliance on LLM-based scoring. However, it can inflate weak model scores, especially in smaller models ($\leq 9B$). (***ii***) **Multi-Binary Evaluation (MBin)**: To better distinguish performance in such cases, we construct binary-choice variants by pairing the correct answer against each distractor independently. A model must select the correct option across all pairs to be marked correct, significantly reducing randomness and more accurately revealing true model capabilities, particularly critical when evaluating across a wide range of model scales. (***iii***) **Chain-of-Thought (CoT) vs. Direct Answering**: In direct evaluation, strict instruction-following is crucial since models are instructed to respond with only the correct option, allowing immediate, format-based extraction without post-processing. In contrast, CoT evaluations encourage models to articulate detailed reasoning steps prior to answering, reflecting human-like problem-solving approaches, and providing leniency regarding response formatting. Here, we use a lightweight state-of-the-art open-source LLM (Qwen-3-4B (Yang et al., 2025a)) in non-thinking mode to extract the final answer. This setup allows us to analyze whether reasoning enhances performance. (***iv***) **Step-wise Reasoning Evaluation**: For CoT responses, we further evaluate the quality of reasoning by comparing model-generated rationales with annotated solution steps. We use the Qwen-3-4B model in its thinking mode for this task, leveraging its strong mathematical reasoning ability to ensure fair and accurate evaluation. The model assigns each response a score between $0$ and $10$, along with a rationale. We further use the rationales to enable error analysis against predefined error categories, providing deep, actionable insights that highlights model limitations and reasoning gaps. We validate the reliability of Qwen-3-4B as the judge model against human scoring and confirm that the rubric accounts for alternative but logically valid reasoning paths; both analyses support the robustness of our evaluation protocol (see Appendix C).

| Models | Size | MCQ | | MBin | | Mathematic Concepts | | | | | | | | | | Duration | | |
|---|---|---|---|---|---|---|---|---|---|---|---|---|---|---|---|---|---|---|
| | | V | +Sub | V | +Sub | GAng | GAre | GLen | Chart | Stat | Arth | Topo | Grph | Cntg | Pzle | Short | Med | Long |
| Random | - | 17.4 | 17.4 | 7.9 | 7.9 | 8.7 | 7.0 | 7.8 | 10.7 | 8.7 | 3.9 | 6.7 | 11.1 | 2.6 | 11.1 | 9.0 | 7.8 | 6.9 |
| Human | - | - | 80.7 | - | 80.7 | 91.3 | 83.1 | 80.4 | 81.3 | 87.0 | 80.8 | 60.0 | 88.9 | 84.2 | 70.4 | 80.3 | 82.1 | 79.6 |
| *Proprietary Models* | | | | | | | | | | | | | | | | | | |
| Claude-3.7-sonnet | - | 26.2 | 27.1 | 8.6 | 9.5 | 17.4 | 9.9 | 5.9 | 8.0 | 17.4 | 11.5 | 13.3 | 5.6 | 5.3 | 9.3 | 8.2 | 11.0 | 9.1 |
| GPT-4o | - | 20.2 | 24.5 | 12.6 | 13.6 | 13.0 | 12.7 | 15.7 | 12.0 | 4.4 | 17.3 | 20.0 | 5.6 | 7.9 | 20.4 | 14.2 | 15.6 | 10.6 |
| Gemini-1.5-Flash | - | 20.5 | 23.1 | 12.6 | 17.6 | 26.1 | 15.5 | 19.6 | 9.3 | 17.4 | 23.1 | 6.7 | 22.2 | 15.8 | 24.1 | 17.9 | 22.1 | 12.1 |
| Gemini-2.0-Flash | - | 28.6 | 31.7 | 14.1 | 20.5 | 30.4 | 23.9 | 27.5 | 13.3 | 8.7 | 19.2 | 13.3 | 16.7 | 7.9 | 33.3 | 25.4 | 24.0 | 11.4 |
| *Open-source Models (< 5B)* | | | | | | | | | | | | | | | | | | |
| Qwen2.5-VL | 3B | 26.9 | 27.6 | 19.3 | 19.6 | 26.1 | 23.9 | 23.5 | 21.3 | 34.8 | 17.3 | 26.7 | 11.1 | 15.8 | 20.4 | 25.4 | 23.4 | 15.9 |
| InternVL2.5 | 2B | 24.3 | 20.7 | 14.3 | 14.5 | 21.7 | 9.9 | 27.5 | 10.7 | 4.4 | 15.4 | 20.0 | 0.0 | 15.8 | 16.7 | 17.9 | 16.9 | 8.3 |
| PLM-LLaMA | 3B | 22.9 | 22.1 | 13.6 | 15.0 | 17.4 | 16.9 | 25.5 | 8.0 | 26.1 | 9.6 | 20.0 | 11.1 | 13.2 | 13.0 | 16.4 | 18.8 | 9.1 |
| InternVL3 | 2B | 22.4 | 23.3 | 18.8 | 16.4 | 21.7 | 16.9 | 17.7 | 17.3 | 30.4 | 15.4 | 20.0 | 22.2 | 13.2 | 5.6 | 18.7 | 14.9 | 15.9 |
| *Open-source Models (< 9B)* | | | | | | | | | | | | | | | | | | |
| PLM-LLaMA | 8B | 22.1 | 23.1 | 16.7 | 14.5 | 13.0 | 11.3 | 17.7 | 13.3 | 17.4 | 17.3 | 20.0 | 11.1 | 10.5 | 16.7 | 16.4 | 14.9 | 12.1 |
| Oryx-1.5 | 7B | 22.6 | 22.6 | 16.9 | 17.4 | 13.0 | 23.9 | 23.5 | 9.3 | 21.7 | 23.1 | 20.0 | 5.6 | 18.4 | 11.1 | 20.2 | 20.8 | 10.6 |
| LLaVA-OV | 7B | 20.7 | 21.2 | 14.8 | 15.5 | 8.7 | 15.5 | 17.7 | 16.0 | 30.4 | 17.3 | 13.3 | 5.6 | 15.8 | 11.1 | 16.4 | 18.8 | 10.6 |
| LongVA-DPO | 7B | 21.4 | 21.7 | 16.2 | 14.1 | 8.7 | 15.5 | 17.7 | 12.0 | 30.4 | 9.6 | 6.7 | 5.6 | 10.5 | 18.5 | 14.9 | 11.7 | 15.9 |
| Video-R1 | 7B | 21.4 | 17.4 | 16.0 | 16.2 | 8.7 | 22.5 | 25.5 | 16.0 | 26.1 | 13.5 | 6.7 | 5.6 | 13.2 | 9.3 | 16.4 | 16.9 | 15.2 |
| InternVL2.5 | 8B | 24.3 | 24.8 | 18.6 | 18.6 | 26.1 | 19.7 | 17.7 | 17.3 | 21.7 | 19.2 | 26.7 | 11.1 | 10.5 | 20.4 | 17.9 | 22.7 | 14.4 |
| LLaVA-Video | 7B | 26.9 | 26.4 | 20.0 | 19.3 | 13.0 | 21.1 | 31.4 | 17.3 | 17.4 | 15.4 | 26.7 | 5.6 | 18.4 | 18.5 | 23.9 | 20.8 | 12.9 |
| InternVideo2.5 | 8B | 25.2 | 28.6 | 19.1 | 19.1 | 34.8 | 22.5 | 15.7 | 14.7 | 21.7 | 19.2 | 20.0 | 27.8 | 10.5 | 18.5 | 18.7 | 22.1 | 15.9 |
| Qwen2.5-VL | 7B | 26.7 | 27.9 | 19.8 | 19.1 | 8.7 | 25.4 | 25.5 | 18.7 | 13.0 | 23.1 | 13.3 | 5.6 | 15.8 | 16.7 | 22.4 | 19.5 | 15.2 |
| InternVL3 | 8B | 29.1 | 27.9 | 20.0 | 20.7 | 13.0 | 29.6 | 27.5 | 13.3 | 13.0 | 28.9 | 20.0 | 22.2 | 15.8 | 14.8 | 25.4 | 24.0 | 12.1 |
| VideoChat-R1 | 7B | 27.6 | 29.1 | 21.2 | 21.2 | 8.7 | 22.5 | 31.4 | 21.3 | 17.4 | 30.8 | 6.7 | 11.1 | 15.8 | 18.5 | 26.9 | 20.1 | 16.7 |
| *Open-source Models (< 40B)* | | | | | | | | | | | | | | | | | | |
| Aria | 34B | 23.8 | 26.4 | 17.4 | 19.1 | 8.7 | 25.4 | 19.6 | 22.7 | 17.4 | 19.2 | 20.0 | 11.1 | 21.1 | 11.1 | 21.6 | 16.9 | 18.9 |
| Oryx-1.5 | 32B | 30.5 | 33.1 | 22.9 | 24.1 | 30.4 | 39.4 | 31.4 | 10.7 | 17.4 | 21.2 | 6.7 | 11.1 | 15.8 | 33.3 | 27.6 | 29.9 | 13.6 |
| Qwen2.5-VL | 32B | 32.4 | 32.6 | 25.7 | 24.8 | 43.5 | 31.0 | 25.5 | 14.7 | 26.1 | 26.9 | 6.7 | 27.8 | 10.5 | 33.3 | 28.4 | 30.5 | 14.4 |
| InternVL2.5 | 38B | 31.0 | 33.6 | 24.1 | 26.0 | 43.5 | 38.0 | 39.2 | 8.0 | 13.0 | 32.7 | 6.7 | 11.1 | 18.4 | 29.6 | 34.3 | 31.8 | 10.6 |
| InternVL3 | 38B | 31.7 | 35.7 | 25.2 | 29.5 | 34.8 | 42.3 | 37.3 | 13.3 | 17.4 | 25.0 | 13.3 | 33.3 | 26.3 | 40.7 | 35.8 | 38.3 | 12.9 |
| *Open-source Models (< 80B)* | | | | | | | | | | | | | | | | | | |
| LLaVA-Video | 72B | 28.3 | 30.0 | 20.2 | 24.3 | 8.7 | 32.4 | 25.5 | 20.0 | 13.0 | 36.5 | 13.3 | 22.2 | 21.1 | 24.1 | 27.6 | 27.3 | 17.4 |
| LLaVA-OV | 72B | 25.5 | 28.3 | 21.0 | 24.8 | 17.4 | 31.0 | 23.5 | 12.0 | 21.7 | 38.5 | 20.0 | 27.8 | 18.4 | 31.5 | 30.6 | 28.6 | 14.4 |
| InternVL2.5 | 78B | 33.3 | 31.7 | 28.3 | 27.9 | 39.1 | 36.6 | 31.4 | 18.7 | 26.1 | 32.7 | 26.7 | 27.8 | 13.2 | 27.8 | 33.6 | 35.1 | 13.6 |
| Qwen2.5-VL | 72B | 36.9 | 37.6 | 26.0 | 27.9 | 26.1 | 36.6 | 31.4 | 17.3 | 30.4 | 38.5 | 20.0 | 16.7 | 18.4 | 29.6 | 34.3 | 29.2 | 19.7 |
| InternVL3 | 78B | 33.3 | 31.7 | 28.3 | 27.9 | 39.1 | 36.6 | 31.4 | 18.7 | 26.1 | 32.7 | 26.7 | 27.8 | 13.2 | 27.8 | 33.6 | 35.1 | 13.6 |

**Table 2:** Performance on VideoMathQA using *direct answer prompting* in both MCQ and MBin settings. We report results with video-only (V) and video+subtitle (+Sub) inputs. MBin performance is further broken down across ten mathematical concepts and three video duration categories. The mathematic concepts are geometric angle (GAng), geometric area (GAre), geometric length (GLen), chart reading (Chart), statistics and probability (Stat), arithmetic and calculus (Arth), topology (Topo), graph theory (Grph), counting (Cntg) and puzzles (Pzle).

**Human Evaluation:** We perform a comprehensive human evaluation using a pool of 8 annotators. Each annotator is assigned a roughly equal subset of questions and is instructed to watch the corresponding video and solve the given question, with a maximum of 20 minutes allowed per question. We allow flexibility in how they watch the video, they may choose to increase playback speed or skip segments at their choice.

## 4.2 QUANTITATIVE ANALYSIS

Tab. 2 presents the direct answer evaluation, and Tab. 3 shows the chain-of-thought (CoT) evaluation. Both tables cover the MCQ and MBin evaluation, with and without subtitles, providing a comprehensive view of model capabilities. Tab. 3 additionally reports CoT step evaluation scores based on alignment with annotated reasoning traces. Proprietary models tend to benefit significantly from CoT prompting, whereas open-source models show mixed gains. In step evaluation, GPT-o4-mini achieves the highest score of 6.9, with Qwen2.5-VL-72B leading among open models with a score of 5.0.

**How does model size influence performance?** Across both MCQ and MBin settings, we observe that *model performance improves with scale*, both with CoT prompting (Tab. 3) and direct answering (Tab. 2). For instance, in the CoT MBin with subtitles, InternVL-3 improves steadily with size: 20.0% (8B) to 25.0% (38B) and 27.9% (72B). Similar trends hold for other models (LLaVA-Video, LLaVA-OneVision, Qwen2.5-VL), indicating that larger models are better at retaining temporal context, focusing on key visual details, and grounding information across modalities, all crucial for video-based mathematical reasoning. *However, scale alone does not determine performance. Smaller, newer models often outperform older, larger ones.* For example, InternVL-3-38B surpasses multiple 72B models (LLaVA-Video-72B, LLaVA-OneVision-72B) in both CoT and direct answers. Newer

| Models | Size | MCQ | | MBin | | Mathematic Concepts | | | | | | | | | | Duration | | | CoT |
|---|---|---|---|---|---|---|---|---|---|---|---|---|---|---|---|---|---|---|---|
| | | V | +Sub | V | +Sub | GAng | GAre | GLen | Chart | Stat | Arth | Topo | Grph | Cntg | Pzle | Short | Med | Long | Eval |
| Random | - | 17.4 | 17.4 | 7.9 | 7.9 | 8.7 | 7.0 | 7.8 | 10.7 | 8.7 | 3.9 | 6.7 | 11.1 | 2.6 | 11.1 | 9.0 | 7.8 | 6.9 | - |
| Human | - | - | 80.7 | - | 80.7 | 91.3 | 83.1 | 80.4 | 81.3 | 87.0 | 80.8 | 60.0 | 88.9 | 84.2 | 70.4 | 80.3 | 82.1 | 79.6 | - |
| *Open-source (< 9B)* | | | | | | | | | | | | | | | | | | | |
| Video-R1 | 7B | 23.8 | 27.6 | 18.1 | 20.0 | 13.0 | 26.8 | 23.5 | 9.3 | 13.0 | 34.6 | 20.0 | 16.7 | 18.4 | 16.7 | 21.6 | 26.0 | 11.4 | 3.9 |
| LLaVA-Video | 7B | 26.4 | 23.6 | 20.0 | 16.0 | 4.4 | 15.5 | 23.5 | 16.0 | 21.7 | 7.7 | 26.7 | 0.0 | 21.1 | 18.5 | 16.4 | 16.9 | 14.4 | 2.7 |
| Qwen2.5-VL | 7B | 25.2 | 29.5 | 17.6 | 18.3 | 13.0 | 15.5 | 11.8 | 20.0 | 21.7 | 36.5 | 13.3 | 16.7 | 10.5 | 16.7 | 16.4 | 20.1 | 18.2 | 3.7 |
| InternVL3 | 8B | 28.8 | 26.9 | 17.9 | 20.0 | 17.4 | 22.5 | 27.5 | 13.3 | 4.4 | 17.3 | 13.3 | 16.7 | 7.9 | 24.1 | 19.4 | 23.4 | 9.9 | 3.4 |
| InternVideo2.5 | 8B | 24.3 | 27.6 | 18.3 | 19.8 | 26.1 | 25.4 | 13.7 | 14.7 | 26.1 | 21.2 | 13.3 | 27.8 | 18.4 | 18.5 | 17.9 | 25.3 | 15.2 | 3.0 |
| VideoChat-R1 | 7B | 22.4 | 28.3 | 21.4 | 19.8 | 13.0 | 29.6 | 15.7 | 10.7 | 17.4 | 26.9 | 6.7 | 22.2 | 18.4 | 24.1 | 20.9 | 25.3 | 12.1 | 3.6 |
| *Open-source (< 40B)* | | | | | | | | | | | | | | | | | | | |
| Oryx-1.5 | 32B | 29.1 | 33.6 | 21.7 | 25.2 | 34.8 | 35.2 | 43.1 | 13.3 | 17.4 | 21.2 | 20.0 | 22.2 | 18.4 | 22.2 | 30.6 | 29.2 | 15.2 | 3.7 |
| InternVL3 | 38B | 30.0 | 31.4 | 21.7 | 25.0 | 43.5 | 31.0 | 25.5 | 16.0 | 17.4 | 32.7 | 20.0 | 11.1 | 13.2 | 31.5 | 26.9 | 31.2 | 15.9 | 4.1 |
| Qwen2.5-VL | 32B | 31.4 | 36.9 | 22.6 | 27.1 | 47.8 | 29.6 | 33.3 | 16.0 | 26.1 | 32.7 | 13.3 | 16.7 | 23.7 | 29.6 | 29.1 | 32.5 | 18.9 | 4.9 |
| *Open-source (< 80B)* | | | | | | | | | | | | | | | | | | | |
| LLaVA-Video | 72B | 23.6 | 29.3 | 14.8 | 18.6 | 8.7 | 22.5 | 17.7 | 14.7 | 8.7 | 21.2 | 26.7 | 11.1 | 26.3 | 20.4 | 17.2 | 21.4 | 16.7 | 3.1 |
| LLaVA-OV | 72B | 23.3 | 26.9 | 14.3 | 18.1 | 8.7 | 14.1 | 19.6 | 13.3 | 21.7 | 26.9 | 20.0 | 22.2 | 10.5 | 25.9 | 15.7 | 23.4 | 14.4 | 3.2 |
| Qwen2.5-VL | 72B | 37.4 | 36.9 | 24.5 | 28.6 | 30.4 | 31.0 | 31.4 | 24.0 | 21.7 | 50.0 | 13.3 | 22.2 | 15.8 | 25.9 | 27.6 | 34.4 | 22.7 | 5.0 |
| InternVL3 | 78B | 34.1 | 37.1 | 25.2 | 27.9 | 39.1 | 39.4 | 33.3 | 13.3 | 26.1 | 23.1 | 33.3 | 22.2 | 10.5 | 40.7 | 28.4 | 36.4 | 17.4 | 4.9 |
| *Proprietary Models* | | | | | | | | | | | | | | | | | | | |
| Claude-3.7-sonnet | - | 24.8 | 29.5 | 12.1 | 19.3 | 34.8 | 29.6 | 19.6 | 4.0 | 26.1 | 13.5 | 20.0 | 16.7 | 21.1 | 22.2 | 23.1 | 26.0 | 7.6 | 4.2 |
| GPT-4o | - | 27.1 | 34.3 | 18.6 | 22.9 | 26.1 | 22.5 | 17.7 | 17.3 | 30.4 | 32.7 | 20.0 | 33.3 | 13.2 | 25.9 | 19.4 | 29.9 | 18.2 | 4.9 |
| Gemini-2.0-Flash | - | 35.2 | 38.8 | 19.5 | 24.8 | 34.8 | 21.1 | 27.5 | 18.7 | 21.7 | 28.9 | 13.3 | 33.3 | 18.4 | 33.3 | 27.6 | 27.9 | 18.2 | 4.7 |
| GPT-o4-mini | - | 49.8 | 61.4 | 42.1 | 44.8 | 43.5 | 49.3 | 45.1 | 40.0 | 65.2 | 63.5 | 20.0 | 72.2 | 23.7 | 31.5 | 45.5 | 44.8 | 42.4 | 6.9 |

**Table 3:** Performance on VideoMathQA using *chain-of-thoughts prompting* in MCQ and Multi-Binary (MBin) settings. We report results with video-only (V) and video+subtitle (+Sub) inputs to highlight the effect of subtitles on multimodal grounding. MBin performance is further broken down across ten mathematical concepts and three video duration categories. The table covers open-source models across multiple size tiers.

models benefit from stronger architectures, improved visual understanding, and better reasoning, enabling them to outperform larger, previously SoTA models.

**How do proprietary models compare to open-source models?** We evaluate 5 proprietary models with CoT prompting and find that Gemini-2.0-Flash and GPT-o4-mini deliver the best performance. GPT-o4-mini achieves the highest overall accuracy, with $44.8\%$ in CoT MBin with subtitles. It performs particularly well in complex reasoning categories such as arithmetic-calculus ($63.5\%$), statistics ($65.2\%$), and geometric area ($49.3\%$), with performance significantly higher than the average of other proprietary and open-source models. These results suggest that its strong performance stems from a better ability to integrate understanding across vision, audio, text, and background subject knowledge, enabling more coherent mathematical reasoning. While proprietary models continue to lead, our results show that *the gap between proprietary and open-source models is narrowing*. Optimized open-source models such as Qwen2.5-VL-72B and InternVL-3-78B outperform several proprietary counterparts, including Claude-3.7-Sonnet, Gemini-2.0-Flash, and GPT-4o.

**How do subtitles influence model performance?** Subtitles consistently enhance model performance across both CoT evaluation (Tab. 3) and direct answering (Tab. 2), especially for larger open-source and proprietary models. However, the impact of subtitles is not uniform: smaller models ($<5B$ and $<9B$) often show minimal or inconsistent gains. In contrast, reasoning-capable models like GPT-o4-mini improve from $42.1\%$ (video-only) to $44.8\%$ (video+sub), and Qwen2.5-VL improves from $24.5\%$ to $28.6\%$ in the CoT MBin. As shown in Fig. 3b, *models with stronger reasoning capabilities benefit more from subtitles, while smaller models struggle to extract meaningful gains from them.* These improvements reflect the ability to integrate fine-grained audio cues with visual frames, a *'needle-in-a-multimodal-haystack'* challenge, where critical information is distributed across modalities, and stronger reasoning models are better equipped to ground these disparate cues into coherent solutions, while others may overlook small but essential verbal cues.

**How does video length and frame sampling affect performance?** We evaluate model performance across short ($<30s$), medium (30s–2min), and long (2min–1hr) videos and observe two distinct trends (see Fig. 3a). *First*, while most models perform reasonably well in short videos, accuracy generally improves in medium-length videos and declines for longer durations. These trends align with the three reasoning challenges in the benchmark. Short videos often correspond to *problem focused* questions, requiring the model to derive a solution, where success hinges on general mathematical competence and the ability to extract key visual or verbal cues. Medium-length videos commonly involve *concept transfer* questions, where the model is first shown a solution or method, then asked to adapt it to a related problem, favoring models that can effectively comprehend instructions. In contrast, long

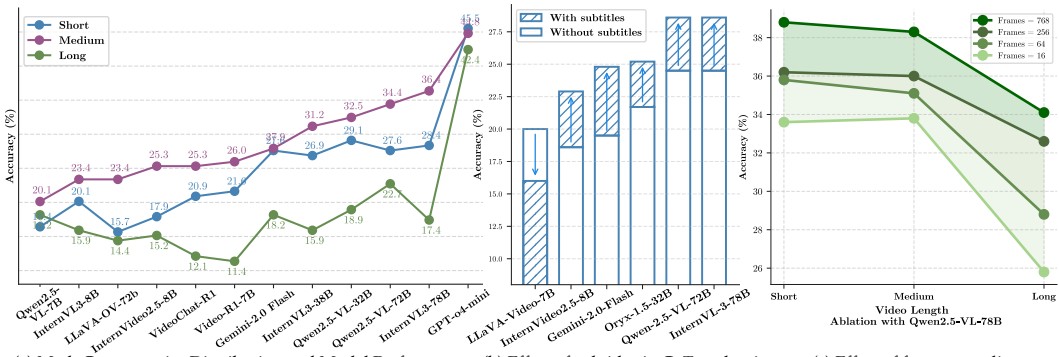

(a) Math Concept-wise Distribution and Model Performance (b) Effect of subtitles in CoT evaluation (c) Effect of frame sampling

**Figure 3:** The figure shows VideoMathQA performance **a)** Across video duration categories under CoT MBin +Sub; **b)** Impact of subtitles under the CoT MBin; and **c)** Effect of varying the number of input frames under CoT MCQ. Overall, models perform best on medium-length videos, and accuracy improves with the inclusion of subtitles and more frames.

videos correspond to *deep instructional comprehension* questions, which require following extended, often non-linear instructional sequences to interpret the context. Here, the information load is higher and the essential clues for solving the problem are scattered over modalities and time. This setting closely aligns with the central challenge, where overlooking even a few critical details can derail the entire reasoning process. *Second*, we study how frame sampling influences performance by evaluating Qwen2.5-VL with 16, 64, 256, and 768-frame settings (see Fig. 3c). We find that *increasing frame count provides consistent improvements, particularly for longer videos*: a 5-point gain for short videos and up to 8 points in long videos, highlighting that models capable of handling extended frame sequences and maintaining long-range temporal coherence are better equipped for video-based mathematical reasoning. Further, in Fig. 4a, we compare video models with vision-blind text only and single-image models, highlighting that in-depth temporal reasoning is required to perform well on VideoMathQAbenchmark.

**How does question difficulty affect performance?** Model accuracy varies notably with question difficulty, while most models solve a moderate proportion of easy questions, they struggle with harder ones (Fig. 4b). GPT-4o answers 96% of easy questions correctly, yet handles only 46% of the hard ones. InternVL-3-78B solves 60% of easy examples, but manages just 8% at the hard level. Other models follow the same trend, exposing a key limitation in current models, their ability to generalize weakens under higher cognitive load.

**Where do models fail in the reasoning process?** Fig. 4c shows a breakdown of model errors from CoT step evaluation across *seven error categories*: problem misunderstanding, failure to

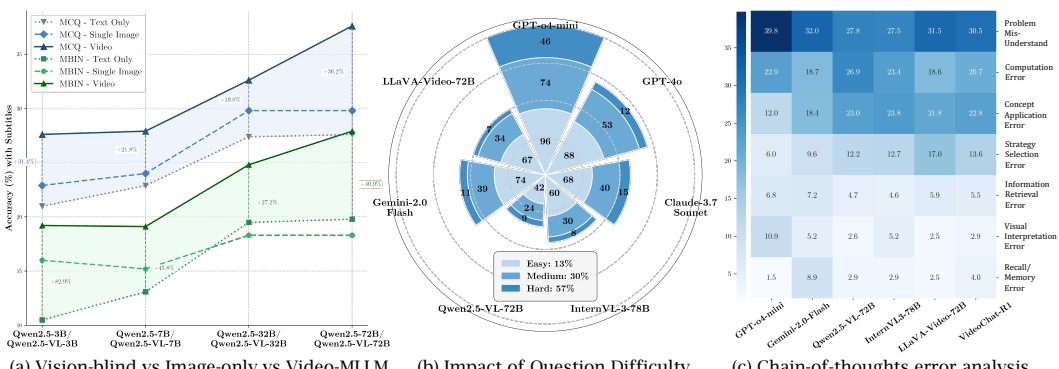

(a) Vision-blind vs Image-only vs Video-MLLM (b) Impact of Question Difficulty (c) Chain-of-thoughts error analysis

**Figure 4:** The figure shows **a)** Comparison among vision-blind, image-only, and video models, highlighting the need for video-level understanding to perform well in VideoMathQA. **b)** Distribution of questions in VideoMathQAacross three difficulty levels for varying reasoning depths, and the relationship between performance and question difficulty across top-performing models. **c)** Error analysis based on CoT step evaluation. Most model errors stem from misunderstanding the question, where models misinterpret what the question asks or overlook critical multimodal cues.

retrieve relevant information, misinterpreting visuals, incorrect concept application, wrong strategy or formula, forgetting previous context, and calculation errors. The most common issue is problem misunderstanding, where models fails to localize the specific information referred to in the question or overlook critical multimodal cues in the video. This reflects the core challenge of our benchmark, where missing even a small verbal or visual detail can derail reasoning entirely. *Proprietary models* like GPT-o4-mini and Gemini-2.0-Flash *show fewer errors in concept application and strategy selection* (12% and 6%), suggesting stronger domain grounding. In contrast, open-source models like InternVL-3 exhibit more broadly distributed errors, with concept application and strategy selection together accounting for 23% of total errors, plus notable computation mistakes. GPT-o4-mini, however, has a higher share of visual interpretation errors, showing difficulty with fine-grained cues like charts and diagrams. See Tab. E.7 for error annotation prompts and category definitions and Appendix A for further discussion on performance across mathematical concepts and human-model comparisons.

## 5 CONCLUSION

In this work, we introduce VideoMathQA, a comprehensive benchmark to evaluate mathematical reasoning in real-world educational videos. It provides curated video–question pairs with step-by-step reasoning across diverse domains and instructional formats, addressing the lack of benchmarks for temporally extended, cross-modal reasoning in multimodal models. Unlike prior benchmarks based on static images and limited context, VideoMathQA requires models to integrate dynamic visual, textual, and auditory cues to solve problems, transfer concepts, and demonstrate deep instructional comprehension.

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

# Appendix

We provide appendix for a deeper understanding and more analysis related to the main paper, arranged as:

1. Additional Quantitative Analysis (Appendix A)
2. Additional Implementation Details (Appendix B)
3. Robustness of Step-wise Reasoning Evaluation (Appendix C)
4. Related Works (Appendix 2)
5. Limitations and Future Work (Appendix D)
6. Prompts (Appendix E)

## A    ADDITIONAL QUANTITATIVE ANALYSIS

**How well do models perform across different mathematical concepts?** We analyze model performances across the ten mathematical categories covered in the benchmark and observe notable variation in their ability to comprehend and solve different mathematical concepts (see Fig. 2a). Current models tend to perform better on questions involving *arithmetic and calculus*, with average accuracy around 32% and GPT-o4-mini achieving the best performance of 63.5% with CoT evaluation. Most models show moderate performance on categories such as *geometric reasoning and puzzles*, with average performance ranging between 24% and 30%. In contrast, *chart reading, topology, graph theory, and statistics and probability* are more challenging for all models. Average accuracy across these categories typically falls between 16% and 21%, with GPT-o4-mini scoring only 20% in topology and graph theory, and a maximum of 40% in chart reading.

**How does human performance compare to model performance?** Humans achieve an overall accuracy of 80.7%, confirming that the benchmark is solvable but challenging. Performance is lower in categories such as topology (60.0%) and puzzle (70.4%), which typically require longer reasoning. Accuracy is also strongly time-dependent: if all questions taking more than 10 minutes are counted as incorrect, overall accuracy drops to 44.3%. This highlights both the difficulty of the benchmark and that extending the 20-minute cap could further improve performance. Compared to evaluated models, humans score about 36 points higher than the best model on average. The gap is smallest in arithmetic and graph theory, but largest in visually demanding categories like topology, counting, and chart Reading, where humans lead by roughly 35-50 points.

## B    ADDITIONAL IMPLEMENTATION DETAILS

For all models, we run direct answering for both MCQ and MBin. For reasoning-capable models, we additionally perform CoT evaluations for both. A lightweight LLM-based post-processing extracts option choices from CoT responses to compute accuracy. Step-wise evaluation is applied only to MCQ CoT responses. Using ground-truth steps, model responses, and step-evaluation critiques, we conduct error analysis for selected models. We also ablate the effect of subtitles across all settings to analyze sensitivity to multimodal input. All prompts used, including those for CoT prompting, postprocessing, step-wise evaluation, error analysis, and subtitle handling are detailed in Appendix E.1 to E.7.

We use 8 A100-80 GB GPUs for all evaluations. For models with $\leq$ 8B parameters, we use data parallelism across 8 workers. For larger models, we utilize tensor parallelism with $TP = 8$. All MLLM evaluations are conducted using lmms_eval (Li et al., 2024a), while evaluations for language-only models are performed using VLLM (Kwon et al., 2023). We provide the implementation of VideoMathQA in lmms_eval, along with scripts to run both MLLM evaluations as part of the submission.

For all evaluated models, we use the recommended number of input frames following the official code base. Specifically, we use 128 frames for Aria (Li et al., 2024c), 512 frames for InternVideo-2.5 (Wang et al., 2025), 64 frames for LLaVA-Video (Zhang et al., 2024b), 128 frames for LongVA (Zhang et al., 2024a), 32 frames for LLaVA-OneVision (Li et al., 2024b) and 768 frames for Qwen2.5-VL (Bai et al., 2025). We use greedy decoding with a temperature of 0 for all MLLM evaluations. For chain of thought (CoT) evaluation, we post-process the responses to extract the final answer option using

Qwen3-4B (Team, 2025) LLM. For the CoT step evaluation, we also use the Qwen3-4B model as the LLM judge and report the average over three runs. Further details on the exact prompts used for CoT prompting, post-processing, step-wise evaluation, error analysis, and subtitle handling are provided in Appendix E.

## C    ROBUSTNESS OF STEP-WISE REASONING EVALUATION

**Validation of Qwen as Judge Model**: To assess the reliability of Qwen-3-4B as a judge model for step-wise reasoning evaluation and its alignment with human scoring, we conduct an experimental analysis. We sample 15 questions from the benchmark and generate reasoning chains for each question using three models: GPT-o4-mini, Qwen-2.5-VL-72B, and InternVL-3-78B, resulting in a total of 45 samples. We then manually assign step-wise reasoning scores to each prediction using the same rubric used during automated evaluation. We compute the average score per model across the 15 questions and compare them

| Evaluator | GPT-o4-mini | Qwen-2.5-VL | InternVL-3 |
|---|---|---|---|
| Qwen-3 | 7.0 | 5.6 | 6.4 |
| Human | 6.2 | 4.7 | 5.3 |

against the scores assigned by Qwen-3. We observe small differences in the absolute scores, since manual ratings are harder to calibrate and can vary in strictness or leniency depending on the sample. As a result, direct comparison is not always meaningful. However, the relative ranking remains consistent (GPT-4 > InternVL-3 > Qwen-2.5-VL). This validates that Qwen-3 provides reliable scoring and aligns well with human evaluation.

**Accounting for Alternative Correct Reasoning Paths**: Our step-wise evaluation rubric explicitly instructs the judge model to assign a full score if the final answer is correct and the reasoning is logically valid, even if the steps differ from the annotated ground truth. This instruction is stated in the evaluation prompt under Scoring Rubric-2, and is consistently used during evaluation of reasoning chains. To examine how well this instruction is followed during step-wise evaluation, we analyze the frequency and scoring of alternative reasoning paths. We focus on the MCQ Chain-of-Thought with Subtitles setting and evaluate the two strongest performing models: Qwen2.5-VL-72B and GPT-o4-mini. For cases where the final answer is correct, we manually examined whether the predicted reasoning matched the annotated steps. For samples where the model took a different reasoning path, we analyse the scores assigned by the judge model.

Qwen2.5-VL-72B answered 148 questions correctly, and among these, 46 followed a different reasoning path, all of which received a full score. GPT-o4-mini correctly answered 256 questions, 110 using alternative reasoning; 109 of these received full scores. We also observe that 22 of these samples were common cases in which both models followed reasoning paths different from the ground truth. These results indicate that the judge model consistently follows the rubric and assigns full credit when the reasoning is logically valid.

The evaluation prompt asks the judge model to produce a structured scorecard that includes a numeric score along with a critique (see Appendix E). We present the LLM judge evaluator generated critique for one such example: "*The prediction uses coordinate geometry to show perpendicularity via dot product, which is a valid alternative reasoning path. While it differs from the ground truth's geometric transformation approach, the steps are logically sound and lead to the correct answer.*" This shows that the evaluator appropriately recognizes and credits valid alternative reasoning.

**Qwen3-4B for Step-wise Reasoning Evaluation**: We chose Qwen3-4B primarily because it is open-source, reproducible, and *accessible for low-resource settings*. Importantly, we use Qwen3-4B in thinking mode for step evaluation and prompt the model to assign a score and provide a justification for the score in the form of a critique. This encourages the evaluation to be more grounded, consistent, and interpretable (see Appendix E.1). To validate robustness, we additionally conduct step-wise evaluations using three larger Qwen3 models (8B, 14B, and 30B-A3) for four representative models: GPT-o4-mini, Gemini-2.0-Flash, Qwen2.5-VL-72B, and InternVL-78B. As shown in the table, the trends are consistent, and the average scores across different Qwen3 model sizes are very close to those reported using Qwen3-4B. We observe that larger

| Model | 4B | 8B | 14B | 30B-A3 | Avg |
|---|---|---|---|---|---|
| GPT-o4-mini | 6.9 | 6.6 | 6.6 | 6.9 | 6.8 |
| Qwen2.5-VL-72B | 5.0 | 4.8 | 4.8 | 5.0 | 4.9 |
| InternVL-78B | 4.9 | 4.7 | 4.6 | 4.9 | 4.8 |
| Gemini-2.0-Flash | 4.7 | 4.5 | 4.4 | 4.6 | 4.6 |

Qwen3 models can be slightly more stringent in evaluation, but the relative performance ranking remains consistent across all model sizes (GPT-o4-mini > Qwen2.5-VL-72B > InternVL-78B > Gemini-2.0-Flash). This demonstrates that Qwen-3-4B provides reliable comparative insights while maintaining reproducibility and computational efficiency.

## D    LIMITATIONS AND FUTURE WORK

VideoMathQA is an initial effort to annotate question-answer pairs along with step-by-step chain-of-thought reasoning in temporally rich videos, where the answer cannot be inferred merely from a few static frames or the audio transcript. The selection and annotation of these samples require a significant amount of time. For example, on average, it took graduate-level experts with at least a Master's degree in Science approximately 30 minutes to find a suitable video, 40 minutes to write a good question and answer, and 1 hour to compose detailed step-by-step reasoning. In total, annotating one sample took around 2 to 2.5 human hours, amounting to approximately 115 working days for 420 samples. This effort is substantial and makes scaling the dataset size difficult. We identify this as a limitation of our work and plan to explore semi-automatic annotation pipelines in the future to reduce the annotation effort while maintaining high annotation quality.

## E    PROMPTS

Below we provide all the LLM prompts used in this work, including (i) CoT prompting to elicit step-by-step reasoning, (ii) post-processing prompts to extract final answers from free-form responses, (iii) evaluation prompts for step-wise reasoning assessment, (iv) error analysis prompts for categorizing mistakes, and (v) subtitle-handling prompts to ensure multimodal alignment.

**LLM Evaluation Prompt for Chain of thoughts Step Evaluation**

You are a intelligent assistant for grading math question solutions. You will be given: A mathematical question (`question`) with multiple-choice options (`options`); A list of numbered ground truth steps (`gt_steps`) showing the correct reasoning to solve a math problem; A answer (`answer`) that is the correct final solution to the question; A model prediction (`prediction`) that includes the steps the model followed and possibly the final answer.

**TASK:** Compare the prediction to gt_steps and assign a score out of 10 using the rubric below. You must reward both matching logic and valid alternative reasoning. Avoid overly strict step-by-step comparison, instead focus if the model follows a coherent and plausible mathematical approach.

**Scoring Rubric:**
1. **Relative Step Matching (Main Criterion)**
   - Count the total number of ground truth steps: N.
   - Evaluate how many predicted steps correctly align with gt_steps in terms of mathematical logic, reasoning, or computations.
   - Score = (matching steps / N) × 10, rounded to nearest whole number.
   - A step MATCHES if it serves the same mathematical purpose, even if phrased or ordered differently.

2. **Correct Final Answer via Different Reasoning**
   - If the model's final answer is correct, and the steps are logically valid (even if they differ from `gt_steps`, assign a full score of 10.
   - Ignore number of matching steps in this case unless the reasoning is clearly flawed or incoherent.
   - Reduce the score proportionally if reasoning contradicts parts of the ground truth.

3. **Implicit or Inferred Steps**
   - Do NOT penalize if early steps are skipped, but later logic clearly depends on them.
   - If a model does not state "identify the chart," but proceeds to use correct values, assume it did this implicitly.
   - ALWAYS check for implied steps before reducing the score.

4. **Ignore Superficial Differences**
   - Do NOT deduct score for formatting, different notation or variable names, or additional clarifications.
   - FOCUS on the mathematical meaning, not the literal step match.

**Output Format**
```
SCORE_CARD: {"matched_steps":  "X/N", "final_answer_correct":
<0 or 1>, "critique":  "<2--3 sentence summary>", "score":
<0--10>}
```

Be strict when awarding credit. Do NOT be lenient. Carefully evaluate how far the model's reasoning aligns with the ground truth steps before assigning a score.

---

**Prompt for LLM Based Post Processing Multiple-Choice Chain-of-Thought Responses**

Given the original multiple-choice options and a model-generated answer containing reasoning and a final answer, identify the option that best matches the final answer and return only the corresponding letter (A, B, C, D, or E).

The options are: {`Options`}

The model response is: {`Response`}

Only return the letter A, B, C, D, or E. If none is found, return `None`.

**Prompt for Multiple-Choice Evaluation with Direct Answering**

Select the best answer to the following multiple-choice question based on the video. Respond with only the letter (A, B, C, D or E) of the correct option.

Answer with the option's letter (A or B) from the given choices directly.

**Prompt for Multiple-Choice Evaluation with Chain-of-Thought**

Select the best answer to the following multiple-choice question based on the video. Respond with the letter (A, B, C, D or E) of the correct option.

First, please perform reasoning, and think step-by-step to provide the best answer to the following question with the option's letter (A, B, C, D or E) from the given choices.

**Prompt for Multiple-Choice Evaluation with Subtitles**

The subtitles of the video are listed below:

{Subtitles}

Select the best answer to the following multiple-choice question based on the video. Respond with only the letter (A, B, C, D or E) of the correct option.

**Prompt for Multi-Binary Evaluation with Direct Answering**

Select the best answer to the following multiple-choice question based on the video. Respond with only the letter (A or B) of the correct option.

Answer with the option's letter (A or B) from the given choices directly.

---

**LLM Evaluation Prompt for Error Analysis using CoT Step Evaluation**

You are an intelligent assistant for analyzing model-generated math solutions. You will be given the following: A mathematical question (`question`) with multiple-choice options (`options`); A list of numbered ground truth steps (`gt_steps`) showing the correct reasoning and a correct final solution (`answer`); A model prediction (`prediction`) that includes the steps the model followed; A critique (`critique`), which is a short rationale describing the quality of the model's reasoning and its alignment with the ground truth.

**TASK:** Your job is to carefully read and analyze the model's prediction. Compare it to the ground truth steps and determine whether the prediction contains any of the following 7 types of reasoning errors. Use the critique and reason about where and why the model diverges from correct logic. Then, assign all relevant error category labels from the list below. The error types are not specific to each step; instead, identify a set of error types that apply to the overall reasoning.

- **TYPE1: Question Misunderstanding Error**: The model fails to understand what the question is asking or misunderstands its demand. It cannot correctly interpret which quantity, relationship, or part of the video is referenced, often mixing up figures, charts, or examples.
- **TYPE2: Information Retrieval Failure**: The model cannot locate or recognize the needed data in the video, including charts. It overlooks presented numbers, labels, angles, side lengths, diagrams, or text overlays, so no raw facts are available for further processing.
- **TYPE3: Visual Interpretation Error**: Although the model attends to the correct visual element, it reads it wrongly, misinterpreting axis scales, bar heights, marker positions, legends, or estimating distances and angles improperly.
- **TYPE4: Concept Application Error** The model knows which principle or method applies but is not able to execute it properly on the given question. It may recall the right concept yet misalign variables, swap parameters, or break the logical steps needed for that specific problem.
- **TYPE5: Strategy & Formula Selection Error**: The model picks an entirely inappropriate approach, choosing the wrong theorem, formula, or problem-solving strategy for the task.
- **TYPE6: Recall & Memory Error**: The model forgets or ignores earlier information that are essential. It also covers cases when it contradicts key information from earlier in the question or in its own reasoning. This includes dropping previously used values, repeating steps unnecessarily, or breaking logical flow by not following through on earlier steps.
- **TYPE7: Computational Error**: The model has the correct inputs and method but makes calculation mistakes, incorrect addition, subtraction, multiplication, division, rounding, or unit-conversion errors.

**RULES:**
- Multiple error types may apply to a single prediction. Errors may be global (relevant to all steps) or local (relevant to one or a few steps).
- Important note: If the model uses different but valid reasoning and arrives at the correct answer, assign: none, even if the steps do not match the ground truth.
- Do not assign an error unless you are confident it reflects a real mistake. Important note: If you are not confident about which error applies, do not guess. Use uncategorized instead of forcing a type.
- If a clear mistake exists, but it does not match any of the listed error types, assign: uncategorized.

**OUTPUT FORMAT:** Return only a comma-separated list of error type labels. Examples: -
`TYPE2, TYPE4`
- `none`
- `uncategorized`
Do not include any explanations or extra text.

---

