# OpenReview forum: "VideoMathQA: Benchmarking Mathematical Reasoning via Multimodal Understanding in Video"
_ICLR.cc/2026/Conference — ICLR 2026 Poster_

### Official Review · Reviewer_oRiV · 2025-10-26

**Soundness:** 4
**Presentation:** 4
**Contribution:** 3
**Rating:** 8
**Confidence:** 4

**Summary:**

This paper introduces VideoMathQA, a new benchmark designed to evaluate mathematical reasoning in videos. The authors argue that this task presents unique challenges not found in static-image or text-based benchmarks, requiring models to interpret fine-grained visual information, read text, and integrate spoken cues that unfold non-linearly over time.

**Strengths:**

1. The effort invested in data quality is a major strength. The three-stage annotation pipeline (Video Selection, QA Annotation, Step-wise Reasoning) is exceptionally rigorous. Using "expert science graduates" and "involving different annotators for verification at each stage" ensures high fidelity. The fact that this process required "115 person-days" demonstrates a significant and laudable effort.
2. The experimental setup is thorough. The paper evaluates a wide range of models using four distinct strategies: standard Multiple-Choice (MCQ), a more robust Multi-Binary (MBin) evaluation, Chain-of-Thought (CoT) vs. direct answering , and a detailed Step-wise Reasoning Evaluation.
3. The analysis provides clear insights into current model limitations including: (a) A large gap between human performance and the best model (b) The error analysis (c) Temporal Analysis and (d) The ablations on subtitles and frame sampling.

**Weaknesses:**

1. The primary limitation, which the authors correctly identify in the appendix, is the size of the dataset (420 video-question pairs). While the quality is extremely high, this small scale limits its utility for training and raises questions about the statistical robustness of the evaluation.
2. The benchmark appears to be English-centric. The paper does not specify the languages covered, but the examples (e.g., in Figure 1) and discussion imply an English-only dataset. This limits its applicability for evaluating mathematical reasoning in other languages, where instructional content and terminology may differ.

**Questions:**

See Weaknesses.

---

> ### Author Response · Authors · 2025-11-21
> **Response to Reviewer oRiV (1/2)**
>
> We thank the reviewer for their thoughtful and encouraging feedback. We greatly appreciate their recognition of our: (i) **rigorous three-stage annotation pipeline**, involving expert annotators and independent verification at each stage; (ii) **thorough experimental setup**, which evaluates diverse models under MCQ, Multi-Binary, CoT, and step-wise reasoning protocols; and (iii) **comprehensive analyses**, including temporal reasoning, subtitle effects, and detailed error categorization. We address the reviewer’s concern below.
>
> ---
>
> >The primary limitation, which the authors correctly identify in the appendix, is the size of the dataset (420 video-question pairs). While the quality is extremely high, this small scale limits its utility for training and raises questions about the statistical robustness of the evaluation.
>
> ### 1. Size of the VideoMathQA Dataset
>
> - While VideoMathQA includes 420 video-question pairs, each sample offers detailed and comprehensive evaluation information. Every question is paired with an instructional video and enriched with detailed step-by-step reasoning traces, multimodal context (including video frames, audio, and subtitles), and assigned mathematical category. This allows not only for final answer evaluation but also for in-depth analysis of reasoning steps, error analysis and benchmarking the ability of models in understanding temporally distributed cues.
>
> - This level of annotation makes each instance substantially more informative than typical static benchmarks. For example, diagnosing whether a model failed due to visual misinterpretation, memory limitations, or conceptual confusion is only possible due to the structured annotations that we provide.
>
> - In addition, several recent benchmarks in the literature adopt similar scales while targeting high-value reasoning tasks. For example, WildVision [1] (500 image-based QA pairs), GTA [2] (229 samples with multimodal inputs), VisIT-Bench [3] (592 instruction-following samples), GAIA [4] (466 real-world assistant queries), and WHOOPS! [5] (500 visual commonsense QA examples). Compared to these, each instance in VideoMathQA offers deeper supervision across time. While the overall number of examples is modest, the depth, complexity and diagnostic value of each sample make the benchmark a valuable resource for evaluating multimodal mathematical reasoning in videos.
>
> - While the size of the dataset is modest, its annotation depth, diversity across 10 mathematical domains, and temporal richness make it a statistically meaningful and diagnostic benchmark for evaluating multimodal mathematical reasoning. Our analyses covering the effect of model size, subtitle, video length, frame sampling, question difficulty, and error categorization, demonstrate consistent and interpretable trends, confirming that reliable insights can be drawn even at this scale.
>
> - Further, the Multi-Binary (MBin) and step-wise reasoning substantially increase statistical reliability. In the MBin setting, we construct variants by pairing the correct answer against each distractor independently (**1,680 QA pairs in total**). A model must select the correct option across all pairs to be marked correct. This design significantly reduces the probability of random success from **20% to 6.2%**, effectively removing the advantage of guessing or partial elimination, and more accurately revealing true model capabilities even at moderate dataset sizes.
> - In the step-wise reasoning evaluation, each model’s chain-of-thought rollout is compared step by step against the **2,945 human-annotated reasoning steps**. For every predicted step, we evaluate correctness and logical alignment with the corresponding human step using a consistent rubric, producing a large set of fine-grained comparison points per question. This step-wise scoring captures deviations in intermediate reasoning (e.g., missing a computation, misreading a visual cue), stabilizes estimates across categories, and provides statistically robust signals beyond final-answer accuracy.
>
> ---
> [1] WildVision: Evaluating Vision-Language Models in the Wild with Human Preferences [Lu et al., NeurIPS 2024]
>
> [2] GTA: A Benchmark for General Tool Agents [Wang et al., NeurIPS 2024]
>
> [3] VisIT-Bench: A Benchmark for Vision-Language Instruction Following Inspired by Real-World Use [Bitton et al., NeurIPS 2023]
>
> [4] GAIA: A Benchmark for General AI Assistants [Mialon et al., arXiv 2023]
>
> [5] Breaking Common Sense: WHOOPS! A Vision-and-Language Benchmark of Synthetic and Compositional Images [Bitton-Guetta et al., ICCV 2023]
>
> ---

---

> > ### Author Response · Authors · 2025-11-21
> > **Response to Reviewer oRiV (2/2)**
> >
> > >The benchmark appears to be English-centric. The paper does not specify the languages covered, but the examples (e.g., in Figure 1) and discussion imply an English-only dataset. This limits its applicability for evaluating mathematical reasoning in other languages, where instructional content and terminology may differ.
> >
> > ### 2. Language Coverage
> >
> > - We acknowledge that the current version of VideoMathQA includes only English-language instructional videos and will make this explicit in the updated paper. This choice was primarily motivated by the abundance and diversity of English instructional material, which allowed us to develop and validate the benchmark effectively in this phase.
> >
> > - The annotation pipeline is language-agnostic, and extending VideoMathQA to other languages would mainly require qualified math or science experts fluent in those languages to ensure accurate reasoning-step annotations. This could be explored in future work, together with scaling the dataset.

---

> > > ### Comment · Reviewer_oRiV · 2025-11-26
> > >
> > > Thanks for the authors' explanations, and I would keep my acceptance recommendation for this paper.

---

### Official Review · Reviewer_bHWm · 2025-10-29

**Soundness:** 3
**Presentation:** 3
**Contribution:** 3
**Rating:** 8
**Confidence:** 3

**Summary:**

This paper introduces VideoMathQA, a benchmark designed to evaluate multimodal mathematical reasoning in real-world videos, addressing the limitations of existing static image/text-based math benchmarks that lack support for temporal extension, dynamic visuals, and cross-modal integration. The benchmark comprises 420 manually curated video-question pairs spanning 10 mathematical domains and video durations from 10 seconds to over 1 hour. Each pair includes expert-annotated multi-step reasoning (2,945 total steps) with timestamps, enabling fine-grained evaluation of intermediate inference.

**Strengths:**

1. VideoMathQA fills a gap by focusing on temporally extended cross-modal reasoning for math, an underexplored area in existing video benchmarks.
2. The benchmark leverages graduate-level experts to create detailed step-wise reasoning with timestamps.
3. The four evaluation strategies address limitations of traditional MCQ and provide nuanced insights.
4. The authors systematically investigate factors impacting performance (model size, video duration, subtitles, frame sampling) and conduct error analysis across 7 categories, offering actionable guidance for model improvement.

**Weaknesses:**

1. With only 420 video-question pairs, the benchmark may lack sufficient diversity to generalize across all real-world math instructional scenarios. The high annotation cost raises concerns about scalability.

**Questions:**

1. Given the high annotation cost, what semi-automatic or crowdsourcing strategies are you exploring to scale VideoMathQA? Could synthetic data or data augmentation techniques be integrated without compromising the "real-world" essence of the benchmark?
2. Models perform poorly on some categories, such as topology and graph theory. Could you provide qualitative examples of why these domains are more challenging for current models?

---

> ### Author Response · Authors · 2025-11-21
> **Response to Reviewer bHWm (1/2)**
>
> We thank the reviewer for their thoughtful and constructive feedback. We appreciate their recognition of:
> (i) **temporally extended, cross-modal mathematical reasoning**, addressing a key gap left by existing static image and text benchmarks;
> (ii) **use of graduate-level experts** to create detailed step-wise reasoning annotations with timestamps, enabling fine-grained evaluation of intermediate inference;
> (iii) **four evaluation strategies** that go beyond traditional MCQ and provide nuanced insights into model reasoning; and
> (iv) **comprehensive analysis** covering model size, video duration, subtitles, frame sampling, and error categorization, which together offer actionable guidance for improving multimodal reasoning models.
>
> ---
>
> >With only 420 video-question pairs, the benchmark may lack sufficient diversity to generalize across all real-world math instructional scenarios. The high annotation cost raises concerns about scalability.
>
> >Given the high annotation cost, what semi-automatic or crowdsourcing strategies are you exploring to scale VideoMathQA? Could synthetic data or data augmentation techniques be integrated without compromising the real-world essence of the benchmark?
>
> ### 1. Scaling VideoMathQA: Annotation Cost and Semi-Automatic Expansion
>
> - Investigating a synthetic or semi-automated data pipeline will indeed be useful for curating a much larger training or validation dataset. However, we sincerely believe that a benchmark intended to evaluate diverse models on deep reasoning and thinking modes for educational math videos warrants an effort grounded in human knowledge and human perception. For this reason, it necessarily involves a high annotation cost.
> - We believe that expanding the benchmark more conservatively, even at the expense of higher annotation cost, remains valuable, as each additional instance meaningfully contributes to its depth and interpretability.
> - At the same time, we agree that exploring automated and semi-automated data pipelines is important for scaling training and validation datasets, where scale plays a critical role. However, for evaluation, we believe there is enduring value in curating a high-quality, human-annotated benchmark.

---

> > ### Author Response · Authors · 2025-11-21
> > **Response to Reviewer bHWm (2/2)**
> >
> > >Models perform poorly on some categories, such as topology and graph theory. Could you provide qualitative examples of why these domains are more challenging for current models?
> >
> > ### 2. Challenges in Specific Mathematical Domains
> >
> > To illustrate why topology and graph theory remain particularly challenging domains, we provide two representative examples capturing their distinct reasoning demands.
> >
> > ### Topology
> > - **Question:** The video shows a rope and ring puzzle, and the task is to arrange the steps in the correct order to free the ring from the rope. **Answer:** Move the ring upward → Pick up the ring from the middle of the rope → Free the ring from the rope.
> >
> > - The challenge is to find a continuous deformation (an isotopy) that transforms the ring and rope system from a linked configuration to an unlinked one, without cutting or allowing the objects to intersect.
> >
> > - Solution: Lifting the ring reveals the rope’s central loop where the linkage occurs. The ring is then guided through the rope’s central opening, representing a continuous deformation that preserves topological properties while reconfiguring the linkage. Once unlinked, the ring slides off smoothly.
> >
> > - This task requires integrated reasoning over spatial geometry to interpret object interactions, topological transformations to apply the underlying mathematical principles, and temporal procedural planning to infer the correct causal order of steps.
> >
> > ### Graph Theory
> > - **Question**: The video presents a network of towns connected by roads, and the task is to determine the shortest travel time for a robot to move from town A to town B. **Answer**: 6 minutes
> >
> > - The challenge is to identify the optimal traversal path that minimizes total edge weight while exploring the evolving state of the graph as distances are updated.
> >
> > - Solution: The solver applies Dijkstra’s algorithm, starting from A and iteratively updating the shortest known travel times to each connected town. As edges are explored, the model must keep track of visited and unvisited nodes, continuously revising path estimates. The final shortest route is determined yields a total travel time of 6 minutes.
> >
> > - This task requires strong perceptual understanding to correctly interpret the network structure, node connections, and edge weights as they appear across frames. It also involves mathematical reasoning through algorithmic graph traversal and dynamic path optimization.
> >
> > - Models tend to struggle in these categories because solving such problems requires temporally grounded, multi-step reasoning that integrates perception, mathematical reasoning, and dynamic state tracking. Unlike arithmetic or visual recognition tasks, these problems demand maintaining structured spatial representations and executing progressive reasoning steps that unfold continuously over time, capabilities with which current multimodal models still struggle.

---

> > > ### Comment · Reviewer_bHWm · 2025-11-26
> > >
> > > Thanks for the authors' response. I tend to maintain this positive score.

---

### Official Review · Reviewer_qm1z · 2025-10-29

**Soundness:** 3
**Presentation:** 4
**Contribution:** 3
**Rating:** 6
**Confidence:** 4

**Summary:**

The paper introduce VideoMathQA, a new benchmark focus on cross-modal reasoning on video modality. This dataset is annotated by raduate-level experts with a long annotation process. It fills the gap of high quality math reasoning dataset in the video modailty scope.

**Strengths:**

1. VideoMathQA tests different reasoning scenarios: Direct Problem Solving, Conceptual Transfer, and Deep Instructional Comprehension, which supports hierarchical evaluation and training of models with varying levels of reasoning.
2. The annotation process uses expert manpower to ensure annotation quality, thereby contributing to the benchmark community that current primarily relies on model auto generation.
3. It supports a comprehensive evaluation to MLLM on Math Reasoning with the four question type: MCQ, MBin, Cot vs Direct, and step-wised.

**Weaknesses:**

1. Data scale: The benchmark only contains 420 videos.
2. Lack the SOTA Proprietary Models' performace: the paper hasn't test GPT-5 and Gemini 2.5-pro
3. The absence of cross-modal ablation experiments makes it difficult to quantify the actual contributions of each modality to mathematical reasoning.
4. The paper use Qwen-3-4B as the judge model for Stepwise Reasoning Evaluation, but this is a relatively small size model. So the model's ability in evaluations is questionable, and it also lacks comparative experiments with human eval to demonstrate its feasibility.

**Questions:**

1. The overall dataset is not large. After distributing 420 videos across 10 categories, some categories contain as few as 17 videos (topology and graph theory each account for 4%). Within a category, videos are further distributed across different lengths. As a result, certain categories have only single-digit counts of videos at specific lengths. Could this introduce bias and error?
2. In Fig2 a), the sum of all proportion is **112%**, how could this happen?

---

> ### Author Response · Authors · 2025-11-21
> **Response to Reviewer qm1z (1/4)**
>
> We thank the reviewer for their thoughtful and detailed feedback. We appreciate their recognition of:
> (i) **Coverage of diverse reasoning scenarios** - Direct Problem Solving, Conceptual Transfer, and Deep Instructional Comprehension, enabling hierarchical evaluation across different reasoning levels;
> (ii) **Expert-driven annotation process**, which ensures high-quality reasoning supervision and contributes to the benchmark community that largely relies on model-generated data; and
> (iii) **Comprehensive evaluation protocol**, spanning MCQ, MBin, CoT vs. Direct Answering, and Step-wise Reasoning Evaluation.
>
> ---
>
> >The overall dataset is not large. After distributing 420 videos across 10 categories, some categories contain as few as 17 videos (topology and graph theory each account for 4\%). Within a category, videos are further distributed across different lengths. As a result, certain categories have only single-digit counts of videos at specific lengths. Could this introduce bias and error?
>
> ### 1. Dataset Balance Across Categories
>
> - We clarify that comparisons across models are conducted on overall performance, where each mathematical category is weighted by its number of samples. This ensures that smaller categories (e.g., topology or graph theory) do not disproportionately influence the reported averages. When analyzing performance across video durations, the grouping is based solely on short, medium, and long videos, independent of mathematical category, and weighted according to the number of samples within each duration group. This makes cross-model comparisons fair and unbiased. The category distributions are instead used only for diagnostic analysis (e.g., where models tend to struggle), and illustration of dataset diversity, not for comparing or ranking models.
>
> - Further, the **Multi-Binary (MBin)** and step-wise reasoning substantially increase statistical reliability. In the MBin setting, we construct variants by pairing the correct answer against each distractor independently (**1,680 QA pairs in total**). A model must select the correct option across all pairs to be marked correct. This design significantly reduces the probability of random success from **20% to 6.2%**, effectively removing the advantage of guessing or partial elimination, and more accurately revealing true model capabilities even at moderate dataset sizes.
> - In the step-wise reasoning evaluation, each model’s chain-of-thought rollout is compared step by step against the **2,945 human-annotated reasoning steps**. For every predicted step, we evaluate correctness and logical alignment with the corresponding human step using a consistent rubric, producing a large set of fine-grained comparison points per question. This step-wise scoring captures deviations in intermediate reasoning (e.g., missing a computation, misreading a visual cue), stabilizes estimates across categories, and provides statistically robust signals beyond final-answer accuracy.
>
> ---
>
> >In Fig2 a), the sum of all proportion is 112\%, how could this happen?
>
> ### 2. Clarification on Figure 2(a) Percentages
>
> - We thank the reviewer for catching this. In Figure 2(a), there is a typographical error in the label for Statistics and Probability, which was mistakenly written as 18\% instead of 6\%. The underlying data and analysis are correct, and we will update the figure and caption in the revised version of the paper. We apologize for this oversight.

---

> > ### Author Response · Authors · 2025-11-21
> > **Response to Reviewer qm1z (2/4)**
> >
> > >Data scale: The benchmark only contains 420 videos.
> >
> > ### 3. Size of VideoMathQA Dataset
> >
> > - While VideoMathQA includes 420 video-question pairs, each sample offers detailed and comprehensive evaluation information. Every question is paired with an instructional video and enriched with detailed step-by-step reasoning traces, multimodal context (including video frames, audio, and subtitles), and assigned mathematical category. This allows not only for final answer evaluation but also for in-depth analysis of reasoning steps, error analysis and benchmarking the ability of models in understanding temporally distributed cues.
> >
> > - This level of annotation makes each instance substantially more informative than typical static benchmarks. For example, diagnosing whether a model failed due to visual misinterpretation, memory limitations, or conceptual confusion is only possible due to the structured annotations that we provide (Error analysis in main paper, Figure 4c).
> >
> > - In addition, several recent benchmarks in the literature adopt similar scales while targeting high-value reasoning tasks. For example, WildVision [1] (500 image-based QA pairs), GTA [2] (229 samples with multimodal inputs), VisIT-Bench [3] (592 instruction-following samples), GAIA [4] (466 real-world assistant queries), and WHOOPS! [5] (500 visual commonsense QA examples). Compared to these, each instance in VideoMathQA offers deeper supervision across time. While the overall number of examples is modest, the depth, complexity and diagnostic value of each sample make the benchmark a valuable resource for evaluating multimodal mathematical reasoning in videos.
> >
> > - While the size of the dataset is modest, its annotation depth, diversity across 10 mathematical domains, and temporal richness make it a statistically meaningful and diagnostic benchmark for evaluating multimodal mathematical reasoning. Our analyses covering the effect of model size, subtitle, video length, frame sampling, question difficulty, and error categorization, demonstrate consistent and interpretable trends, confirming that reliable insights can be drawn even at this scale.
> >
> > ---
> >
> > >Lack the SOTA Proprietary Models' performance: the paper hasn't test GPT-5 and Gemini 2.5-pro
> >
> > ### 4. Proprietary Model performance
> >
> > - We thank the reviewer for this valuable suggestion. In response, we have extended our evaluation to include Gemini 2.5-Pro, which natively supports video reasoning. Gemini 2.5-Pro achieves a substantial performance gain with an average accuracy of 65.0\%, outperforming prior models across most categories, particularly those involving fine-grained visual reasoning and temporal comprehension.
> >
> > |Method|Size|Avg|GAng|GAre|GLen|Chart|Stat|Arth|Topo|Grph|Cntg|Pzle|Short|Med|Long|
> > |---|:---:|:---:|:---:|:---:|:---:|:---:|:---:|:---:|:---:|:---:|:---:|:---:|:---:|:---:|:---:|
> > |**Qwen2.5-VL**|72B|28.6|30.4|31.0|31.4|24.0|21.7|50.0|13.3|22.2|15.8|25.9|27.6|34.4|22.7|
> > |**InternVL3**|78B|27.9|39.1|39.4|33.3|13.3|26.1|23.1|33.3|22.2|10.5|40.7|28.4|36.4|17.4|
> > |**Gemini-2.0-Flash**|-|24.8|34.8|21.1|27.5|18.7|21.7|28.9|13.3|33.3|18.4|33.3|27.6|27.9|18.2|
> > |**GPT-o4-mini**|-|44.8|43.5|49.3|45.1|40.0|65.2|63.5|20.0|72.2|23.7|31.5|45.5|44.8|42.4|
> > |**Gemini-2.5-Pro**|-|**65.0**|87.0|77.5|78.4|52.0|78.3|82.7|13.3|66.7|42.1|51.9|72.4|71.4|50.0|
> >
> > ---
> > [1] WildVision: Evaluating Vision-Language Models in the Wild with Human Preferences [Lu et al., NeurIPS 2024]
> >
> > [2] GTA: A Benchmark for General Tool Agents [Wang et al., NeurIPS 2024]
> >
> > [3] VisIT-Bench: A Benchmark for Vision-Language Instruction Following Inspired by Real-World Use [Bitton et al., NeurIPS 2023]
> >
> > [4] GAIA: A Benchmark for General AI Assistants [Mialon et al., arXiv 2023]
> >
> > [5] Breaking Common Sense: WHOOPS! A Vision-and-Language Benchmark of Synthetic and Compositional Images [Bitton-Guetta et al., ICCV 2023]
> >
> > ---

---

> > > ### Author Response · Authors · 2025-11-21
> > > **Response to Reviewer qm1z (3/4)**
> > >
> > > >The absence of cross-modal ablation experiments makes it difficult to quantify the actual contributions of each modality to mathematical reasoning.
> > >
> > > ### 5. Cross-modal Ablations
> > >
> > > - We thank the reviewer for this valuable comment. We note that a subset of cross-modal ablations are included in the main paper to quantify the relative contribution of different modalities. Specifically, Figure 4(a) compares vision-blind (text-only), image-only, and video-MLLM settings, while Table 2 reports the effect of video + subtitles to assess how access to richer temporal and visual context affects reasoning. We did not include a dedicated audio-only ablation in the main paper because most evaluated models do not natively process audio, limiting a consistent comparison across architectures.
> > >
> > > - To ensure a consistent comparison within the same architecture, we compute an additional ablation using Qwen-2.5-VL-72B, evaluating the model across text, image, video, and video + subtitles settings.
> > >
> > >     | Modality | Text | Image | Vid | Vid + Sub |
> > >     |:--|:--:|:--:|:--:|:--:|
> > >     | **Qwen-2.5VL-72B** | 9.3 | 12.6 | 24.5 | 28.6 |
> > >
> > > - We further extend this analysis to a model that supports audio reasoning, and compare text, image, video, video + subtitles, and video + audio inputs using Gemini-2.5-Pro.
> > >     | Modality | Text | Image | Vid | Vid + Sub | Vid + Aud |
> > >     |:--|:--:|:--:|:--:|:--:|:--:|
> > >     | **Gemini-2.5-Pro** | 12.1 | 16.0 | 56.4 | 60.2 | 65.0 |
> > >
> > > - We observe that performance improves consistently as additional modalities are included, with audio providing complementary cues that enhance temporal grounding and alignment with visual content. These results reaffirm that the benchmark effectively captures how access to multiple modalities contributes to improved mathematical reasoning in instructional videos.

---

> > > > ### Author Response · Authors · 2025-11-21
> > > > **Response to Reviewer qm1z (4/4)**
> > > >
> > > > >The paper use Qwen-3-4B as the judge model for Stepwise Reasoning Evaluation, but this is a relatively small size model. So the model's ability in evaluations is questionable, and it also lacks comparative experiments with human eval to demonstrate its feasibility.
> > > >
> > > > ### 6. Validation of Qwen-3-4B as Judge Model
> > > >
> > > > - As discussed in Appendix C (Robustness of Step-wise Reasoning Evaluation), we validate the use of Qwen-3-4B as a judge model through two complementary analyses: **(1)** Comparison with human evaluation, and **(2)** Cross-model consistency analysis across larger Qwen-3 variants. We summarize these findings below for clarity.
> > > >
> > > > **1. Comparison with Human Evaluation**
> > > > - To assess the reliability of Qwen-3-4B as a judge model for step-wise reasoning evaluation and its alignment with human scoring, we conduct an experimental analysis as suggested. We sample 15 questions from the benchmark and generate reasoning chains for each question using three models: GPT-o4-mini, Qwen-2.5-VL-72B, and InternVL-3-78B, resulting in a total of 45 samples. We then manually assign step-wise reasoning scores to each prediction using the same rubric used during automated evaluation. We compute the average score per model across the 15 questions and compare them against the scores assigned by Qwen-3. The results are as follows:
> > > >
> > > >
> > > >     |Evaluator|GPT-o4-mini|Qwen-2.5-VL|InternVL-3|
> > > >     |:--|:--:|:--:|:--:|
> > > >     |**Qwen-3**|7.0|5.6|6.4|
> > > >     |**Human**|6.2|4.7|5.3|
> > > >
> > > >
> > > > - We observe small differences in the absolute scores, since manual ratings are harder to calibrate and can vary in strictness or leniency depending on the sample. As a result, direct comparison is not always meaningful. However, the relative ranking remains consistent (GPT-o4-mini > InternVL-3 > Qwen-2.5-VL). This validates that our choice of Qwen-3 provides reliable scoring and aligns well with human evaluation.
> > > >
> > > > **2. Cross-Model Consistency Across Qwen Variants**
> > > > - Further, to validate robustness, we additionally conduct step-wise evaluations using three larger Qwen3 models (8B, 14B, and 30B-A3) for four representative models: GPT-o4-mini, Gemini-2.0-Flash, Qwen2.5-VL-72B, and InternVL-78B. As shown below, the trends are consistent, and the average scores across different Qwen3 model sizes are very close to those reported using Qwen3-4B.
> > > >
> > > > - Notably, we observe that larger Qwen3 models can get slightly more stringent on evaluation, but the relative performance ranking remains consistent across all model sizes (GPT-o4-mini > Qwen2.5VL-72B > InternVL-78B > Gemini-2.0-Flash). This validates that our choice of Qwen3-4B provides reliable comparative insights while maintaining reproducibility and computational efficiency.
> > > >
> > > >     |Model|4B|8B|14B|30B-A3|Avg|
> > > >     |:--|:--:|:--:|:--:|:--:|:--:|
> > > >     |**GPT-o4-mini**|6.9|6.6|6.6|6.9|6.8|
> > > >     |**Qwen2.5VL-72B**|5.0|4.8|4.8|5.0|4.9|
> > > >     |**InternVL-78B**|4.9|4.7|4.6|4.9|4.8|
> > > >     |**Gemini-2.0-Flash**|4.7|4.5|4.4|4.6|4.6|
> > > >
> > > > - We chose Qwen3-4B primarily because it is open-source, reproducible, and accessible for low-resource settings. Importantly, we use Qwen3-4B in thinking mode for step evaluation and prompt the model to assign a score and provide a justification for the score in the form of a critique. This encourages the evaluation to be more grounded, consistent, and interpretable (see Appendix D, Line 494, scorecard with critique).

---

> > > > ### Comment · Reviewer_qm1z · 2025-11-21
> > > > **Response to the Authors**
> > > >
> > > > Great! You have addressed my concerns and I have increased my score to 8 for the recommended acceptance.

---

### Official Review · Reviewer_fxx1 · 2025-11-01

**Soundness:** 3
**Presentation:** 3
**Contribution:** 3
**Rating:** 4
**Confidence:** 3

**Summary:**

VideoMathQA introduces a benchmark for math problem solving in instructional videos, bridging traditional text/image-based QA and full multimodal video reasoning. It includes a few thousand curated QA pairs from formula-rich YouTube videos, each with aligned frames, audio, transcripts, and formulas across 10 math domains (geometry, calculus, statistics, etc.). Each question features step-by-step solutions (chain-of-thought) for detailed evaluation. The benchmark targets three scenarios: Direct Problem Solving, Conceptual Transfer, and Deep Instructional Comprehension. Baselines with models like Math-LLaVA, MathBLIP, and Video-CoT reveal a large gap from human performance, underscoring the difficulty of integrating visual, textual, and temporal reasoning in mathematical contexts.

**Strengths:**

1. Introduces the first benchmark for math reasoning in instructional videos, capturing dynamic visual and spoken content absent in prior static text/image datasets.

2. Built through 920+ hours of expert annotation, covering 420 problems (~4.5K QA pairs) with aligned formulas, video timestamps, and full step-by-step solutions.

3. Spans 10 math domains and varied video styles (lectures, tutorials, handwritten, slides), testing both short-term perception and long-range reasoning.

4. Measures both answer accuracy and chain-of-thought alignment, includes conditions with/without subtitles, and offers detailed error categorization for model failures.

5. Evaluates multiple vision-language models (e.g., Qwen-VL, InternVL, Math-LLaVA), showing large performance gaps to human accuracy, confirming the benchmark’s difficulty and diagnostic value.

**Weaknesses:**

1. The five-option format simplifies scoring but allows guessing or elimination, limiting evaluation of free-form reasoning and creativity.
2. Some Conceptual Transfer questions may be solvable without watching the video, letting models rely on prior knowledge rather than true video understanding.

**Questions:**

1. Is there any analysis of the model performance without any video and only given the question?
2. Is there any analysis on how reliable the evaluation scores are? is the model following the video at all, or having the video distracting the model from the correct solution?

---

> ### Author Response · Authors · 2025-11-21
> **Response to Reviewer fxx1**
>
> We thank the reviewer for their positive and constructive feedback. We appreciate their recognition of: (i) significance of VideoMathQA as the **first benchmark for mathematical reasoning** in instructional videos; (ii) **comprehensive scope of the dataset**, covering 10 mathematical domains, diverse video styles, and multi-stage reasoning supervision; (iii) **strength of our evaluation framework**.
>
> ---
> >Is there any analysis of the model performance without any video and only given the question?
>
> >Is there any analysis on how reliable the evaluation scores are? is the model following the video at all, or having the video distracting the model from the correct solution?
> ### 1. Model Performance Without Video Input
> - The reviewer asks whether models can solve the questions without watching the video, relying only on textual input. This analysis is already included in our ablation study (Figure 4a, Figure 3c). Our experiments clearly demonstrate that VideoMathQA tasks require access to temporal information spread across video frames. This is demonstrated by two ablations.
> - **Text vs Video**: As shown in Figure 4a, we compare video models with vision-blind text-only setting, which directly corresponds to this question. For this comparison, we use Qwen2.5 LLM(only question, no video) and Qwen2.5-VL with full video input. Across all parameter scales (3B, 7B, 32B, 72B), video-based models outperform the text-only counterparts. At the 72B scale, for example, the CoT MCQ accuracy improves from **27.4\% (text-only) to 37.4\% (video-based)**. This corresponds to a 36\% relative improvement when using the video setting compared to the text-only, highlighting that models need to interpret dynamic visual information across time, not just using background or world knowledge alone.
> - **Attention to Video Content**: To further assess model sensitivity to video information, in Figure 3c, we evaluate Qwen2.5-VL-72B under different frame sampling rates (16, 64, 256, 768 frames). We observe that increasing the number of input frames leads to consistent gains, up to 5 points for short and 8 points for long videos. This trend indicates that models benefit from access to richer temporal context and do not treat additional frames as noise or distraction.
>
> ---
>
> >The five-option format simplifies scoring but allows guessing or elimination, limiting evaluation of free-form reasoning and creativity.
>
> ### 2. Multiple-Choice Format and Guessing Advantage
> - The reviewer notes that the five-option format may simplify scoring and allow for guessing or elimination. VideoMathQA addresses this concern through the Multi-Binary (MBin) evaluation format (Table 1 and 2 MCQ vs MBin), which systematically removes the randomness factor. In the MBin setting, we construct variants by pairing the correct answer against each distractor independently (1,680 QA pairs in total). A model must select the correct option across all pairs to be marked correct. This design significantly reduces the probability of random success from **20% to 6.2%**, effectively removing the advantage of guessing or partial elimination, and more accurately revealing true model capabilities.
> ---
> >Some Conceptual Transfer questions may be solvable without watching the video, letting models rely on prior knowledge rather than true video understanding.
> ### 3. Concept Transfer Questions and Reliance on Prior Knowledge
> - Concept transfer questions in VideoMathQA are explicitly designed to require engagement with the instructional content presented in the video before solving the target problem. These questions assess whether the model can generalize a demonstrated mathematical concept to a new but related scenario, rather than relying on prior or world knowledge.
> - Specifically, to answer a concept transfer question, a model must: (i) Identify the demonstration shown in the video (e.g., a worked example, derivation, or illustrated procedure). (ii) Interpret the key mathematical method demonstrated. (iii) Distinguish the new question to be solved from what is demonstrated, identifying the variation in structure or parameters, the model must understand from the video which problem it is required to solve, as the new question often **closely resembles the demonstrated one** and cannot be resolved without attending to the visual and temporal context. (iv) Apply the demonstrated method appropriately to this modified setting to reach the correct solution.
> - For example, a question from our benchmark, such as: “If we changed the angle on the bottom-left from 125° to 110°, what will be the value of X?” Solving this requires identifying from the video which angle was referenced in the demonstration, understanding how it was used, and adapting the reasoning accordingly. The context in which this angle is introduced and used in the demonstration is critical; interpreting it correctly requires temporal grounding and a precise understanding of the video, not merely recalling prior knowledge.

---

> ### Author Response · Authors · 2025-11-27
> **Follow-Up Response to Reviewer fxx1**
>
> Dear Reviewer fxx1,
>
> We hope this message finds you well. We sincerely appreciate the time you have dedicated to reviewing our manuscript. In our response, we have carefully addressed all of your concerns regarding model performance without video input, multiple-choice format and guessing advantage, and concept transfer questions and reliance on prior knowledge.
>
> We highly value your insights for any further feedback you may be willing to share. If you have any additional questions or concerns regarding our responses, we would be happy to clarify them. Your comments are important to us. Thank you again for taking the time to review our work.

---

### Official Review · Reviewer_etW5 · 2025-11-01

**Soundness:** 2
**Presentation:** 3
**Contribution:** 1
**Rating:** 2
**Confidence:** 5

**Summary:**

This paper proposes a benchmark called videoMathQA which contains math/reasoning questions about materials presented in a video. The author claims that this work provides realistic reasoning challenges and they highlight that they use 920 hours of human labour for the annotation.

**Strengths:**

1. The presentation of the paper is clear and the paper writing is good.

2. The human labels might be useful as a point of comparison.

**Weaknesses:**

1. __Limited technical contribution__:

1.1. While the authors claiming that they have filled the gap of video-based reasoning in math or very specific domains, there is work in [1] which have (1). videos in math, biology and various other scientific domains; and (2). question with high-quality human annotations which include reasoning. Therefore, the contribution might be incremental. A discussion and comparison should be made.

[1] Sun et al. "video-SALMONN-o1: Reasoning-enhanced Audio-visual Large Language Model". ICML 2025.

1.2. The size of the dataset is questionable especially when there are only 420 MCQs. The authors should justify whether 420 questions are enough to draw statistically significant conclusions when comparing different models. Especially when 10 sub-categories are given, as shown in Table 1, there are a lot of systems having the same score because there are not enough data samples. This makes analysis in the results part unconvincing because the difference might just be getting a couple more questions right. \
I would encourage the authors to theoretically compute the number of samples needed to have statistically significance in their results.

1.3. Statistics and verification of human annotator quality missing. Statistics and evidence should be provided to show whether the annotators are reliable, such as inter-annotator agreement for their reasoning steps etc. I do not see a reason why we need the number of human-hour since the efficiency of human annotators can vary a lot.

2. __Lack of justification for the need of Human Reasoning__:

While a lot of efforts have been made to human reasoning annotation, it is unclear whether those are helpful performance indicators. While the authors try to do a comparison using LLM-as-a-judge. I have the following concerns regarding this part:
  - I do not understand the results stated in Appendix C (Note that without referring to Appendices, the justification for this is missing in the main paper). Proper metrics, e.g. Pearson Correlation Coefficient at sample level should be given instead of saying the performance metrics are close to each other. Per-sample PCC is suitable here because we are assigning scores and comparing the absolute values of the scores.
  - An easier and more objective way would be to perform rollout and check the rate of correctness to assign scores to each reasoning step. This is standard in reasoning literature and is more controllable and convincing to me than using LLM-as-a-judge, especially with such a small LLM. I wonder why the authors did not try this.

**Questions:**

See weaknesses

---

> ### Author Response · Authors · 2025-11-21
> **Response to Reviewer etW5 (1/4)**
>
> We thank the reviewer for their feedback and address the comments below.
>
> >While the authors claiming that they have filled the gap of video-based reasoning in math or very specific domains, there is work in video-SALMONN-o1 which have (1). videos in math, biology and various other scientific domains; and (2). question with high-quality human annotations which include reasoning. Therefore, the contribution might be incremental. A discussion and comparison should be made.
>
> ### 1. Comparison with Video-SALMONN-o1
> - We appreciate the reviewer’s pointer to video-SALMONN-o1 (Sun et al., ICML 2025). While both benchmarks involve video-based reasoning, their objectives and design principles **differ substantially**. RivaBench focuses on breadth across three domains (academic lectures, stand-up comedy, and synthetic video detection) to evaluate general audio-visual reasoning. In contrast, VideoMathQA specifically targets **depth within the mathematics domain** that demands precise quantitative reasoning. It spans 10 mathematical categories, 3 question types (problem-focused, concept-transfer, and deep-instructional comprehension), and 3 video duration ranges, ensuring that each sample requires detailed and temporally grounded reasoning.
>
> - The academic subset of video-SALMONN-o1 is derived from the M3AV corpus (Chen et al., ACL 2024), which primarily consists of **slide-based** lecture (mean duration of 47.2s). The released benchmark (HuggingFace: tsinghua-ee/RivaBench) contains only question-option-answer triplets, **without any reasoning traces or temporal annotations**. Moreover, no domain-specific analyses, such as math-focused performance trends or sub-category breakdowns, are provided. In contrast, VideoMathQA specifically selects **dynamic instructional videos** that involve evolving visual, textual, and procedural cues over time, such that questions cannot be answered from static context. Each sample thus requires understanding visual transformations, tracking multimodal cues, and reasoning over temporally distributed information. VideoMathQA includes 420 video-question pairs with 2,945 expert-annotated reasoning steps, covering short (10s) to long (1h) instructional videos (mean duration of **241s**/video). All videos and reasoning annotations will be publicly released for full reproducibility and diagnostic evaluation.
>
> - Importantly, the benchmark is constructed to facilitate detailed analysis of model behavior and to surface reasoning weaknesses.
> In this regard, VideoMathQA offers a high-quality, analysis-driven benchmark. Our **comprehensive evaluations**, covering the effects of model size, subtitles, video length, frame sampling, question difficulty, and error categorization, demonstrate consistent and interpretable trends across models. None of these diagnostic analyses are conducted in video-SALMONN-o1, particularly within mathematical reasoning domain, making VideoMathQA the first to provide a systematic, fine-grained understanding of model limitations in video-based mathematics reasoning.

---

> > ### Author Response · Authors · 2025-11-21
> > **Response to Reviewer etW5 (2/4)**
> >
> > >The size of the dataset is questionable especially when there are only 420 MCQs. The authors should justify whether 420 questions are enough to draw statistically significant conclusions when comparing different models. Especially when 10 sub-categories are given, as shown in Table 1, there are a lot of systems having the same score because there are not enough data samples. This makes analysis in the results part unconvincing because the difference might just be getting a couple more questions right. I would encourage the authors to theoretically compute the number of samples needed to have statistically significance in their results.
> >
> > ### 2. Dataset Size and Statistical Significance of Results
> > ### 2.1 Dataset Size
> > - While VideoMathQA includes 420 video-question pairs, each sample offers detailed and comprehensive evaluation information. Every question is paired with an instructional video and enriched with detailed step-by-step reasoning traces, multimodal context (including video frames, audio, and subtitles), and assigned a mathematical category. This enables evaluation beyond final answers, including detailed analysis of reasoning failures, error types, and model behavior over temporally distributed cues.
> >
> > - The structured annotations make each sample considerably more informative compared to existing other benchmarks. They allow us to determine whether a model's error stems from visual misinterpretation, missing an intermediate step, memory limitations, or conceptual confusion: analyses that would not be possible with standard MCQ-only settings (error analysis in main paper, Figure 4c).
> >
> > - Benchmarks targeting high-value reasoning tasks in recent literature operate at a similar scale as ours: WildVision [1] (500 image-based QA pairs), GTA [2] (229 samples with multimodal inputs), VisIT-Bench [3] (592 instruction-following samples), GAIA [4] (466 real-world assistant queries), and WHOOPS! [5] (500 visual commonsense QA examples). Compared to these, each instance in VideoMathQA offers deeper supervision across time. While the overall number of examples is modest, the depth, granularity and diagnostic value of each sample make the benchmark a valuable resource for evaluating multimodal mathematical reasoning in videos.
> >
> > ### 2.2 Statistical robustness
> > To address concerns about statistical significance, we incorporate two mechanisms that substantially increase effective evaluation resolution.
> > - **Multi-Binary (MBin) evaluation:** In the MBin setting, we construct variants by pairing the correct answer against each distractor independently (**1,680 QA pairs in total**). A model must select the correct option across all pairs to be marked correct. This design significantly reduces the probability of random success from 20\% to 6.2\%, effectively removing the advantage of guessing or partial elimination, and more accurately revealing true model capabilities even at moderate dataset sizes.
> > - **Step-wise reasoning evaluation:** Each model’s chain-of-thought rollout is compared step by step against the **2,945 human-annotated reasoning steps**. For every predicted step, we evaluate correctness and logical alignment with the corresponding human step using a consistent rubric, producing a large set of fine-grained comparison points per question. This step-wise scoring captures deviations in intermediate reasoning (e.g., missing a computation, misreading a visual cue), stabilizes estimates across categories, and provides statistically robust signals beyond final-answer accuracy.
> >
> > ---
> > [1] WildVision: Evaluating Vision-Language Models in the Wild with Human Preferences [Lu et al., NeurIPS 2024]
> >
> > [2] GTA: A Benchmark for General Tool Agents [Wang et al., NeurIPS 2024]
> >
> > [3] VisIT-Bench: A Benchmark for Vision-Language Instruction Following Inspired by Real-World Use [Bitton et al., NeurIPS 2023]
> >
> > [4] GAIA: A Benchmark for General AI Assistants [Mialon et al., arXiv 2023]
> >
> > [5] Breaking Common Sense: WHOOPS! A Vision-and-Language Benchmark of Synthetic and Compositional Images [Bitton-Guetta et al., ICCV 2023]
> >
> > ---

---

> > > ### Author Response · Authors · 2025-11-21
> > > **Response to Reviewer etW5 (3/4)**
> > >
> > > >Statistics and verification of human annotator quality missing. Statistics and evidence should be provided to show whether the annotators are reliable, such as inter-annotator agreement for their reasoning steps etc. I do not see a reason why we need the number of human-hour since the efficiency of human annotators can vary a lot.
> > >
> > >
> > > ### 3. Annotator Quality and Verification
> > > - We do not rely on traditional inter-annotator agreement metrics, where two annotators independently label the same sample and their overlap is measured. This approach is less informative for reasoning benchmarks, where multiple valid reasoning paths may exist, which also introduces additional effort in deciding which reasoning trace to prefer.
> > > To avoid this, we adopt a two-stage verification process, where one annotator first generates the **detailed reasoning steps** with careful attention to completeness and accuracy, and a second expert independently verifies the same reasoning chains and refines them for correctness and coherence. By verifying the same reasoning chains, we obtain a more direct and interpretable estimation of inter-annotator agreement. As already discussed in the paper (Line 187), we report the revision rate as an empirical measure of this consistency: 788 out of 2,945 annotated reasoning steps were revised during verification.
> > > - We report the total annotation effort in human-hours to reflect the scale and depth of expert involvement. Unlike crowdsourced labeling, our annotations are produced by graduate-level science experts, each contributing substantial time per video to ensure high-quality. Reporting human-hours thus conveys the true annotation cost and quality assurance effort, a strength also acknowledged by reviewers **fxx1, qm1z, and oRiV**.

---

> > > > ### Author Response · Authors · 2025-11-21
> > > > **Response to Reviewer etW5 (4/4)**
> > > >
> > > > >While a lot of efforts have been made to human reasoning annotation, it is unclear whether those are helpful performance indicators. While the authors try to do a comparison using LLM-as-a-judge. I have the following concerns regarding this part:
> > > >
> > > > >I do not understand the results stated in Appendix C (Note that without referring to Appendices, the justification for this is missing in the main paper). Proper metrics, e.g. Pearson Correlation Coefficient at sample level should be given instead of saying the performance metrics are close to each other. Per-sample PCC is suitable here because we are assigning scores and comparing the absolute values of the scores.
> > > >
> > > > >An easier and more objective way would be to perform rollout and check the rate of correctness to assign scores to each reasoning step. This is standard in reasoning literature and is more controllable and convincing to me than using LLM-as-a-judge, especially with such a small LLM. I wonder why the authors did not try this.
> > > >
> > > > ### 4. Justification for Human Reasoning Annotations and LLM-as-a-Judge Evaluation
> > > >
> > > > We provide additional quantitative analyses to clarify and strengthen the justification for using Qwen-3-4B as the judge model for step-wise reasoning evaluation.
> > > > - **Human Alignment**: To assess the reliability of Qwen-3-4B as a judge model for step-wise reasoning evaluation and its alignment with human scoring, we conduct an empirical analysis as suggested and compute the Pearson Correlation Coefficient (PCC). We randomly sample 50 questions from the benchmark and generate reasoning chains using Qwen-2.5-VL-72B (as opposed to 15 samples in Appendix C to ensure statistically significant results). Human experts then manually assign step-wise reasoning scores to each prediction following the same rubric used during automated evaluation. We compute the average score across the samples and compare them against the scores assigned by Qwen-3-4B. As shown in table below, we observe strong correlation (PCC = 0.78) demonstrating that Qwen-3-4B aligns closely with human judgment, which confirms its reliability as a consistent and interpretable evaluator for step-wise reasoning quality.
> > > >
> > > >     |Evaluator|Qwen-2.5-VL|
> > > >     |---|:---:|
> > > >     |Qwen-3|4.4|
> > > >     |Human|3.7|
> > > >     |PCC (Human, Qwen-3)|0.78|
> > > >
> > > > - **Cross-Model Consistency**: To further verify the robustness of the judge model, we evaluate the score consistency of Qwen-3 across different scales (4B, 8B, 14B, 30B-A3) using the reasoning outputs of Qwen-2.5-VL-72B. The average step-wise reasoning scores across 420 samples and the corresponding Pearson Correlation Coefficients (PCC) between Qwen-3-4B and its larger variants are shown below:
> > > >
> > > >
> > > >     |Model |Scores|PCC (vs 4B)|
> > > >     |---|:---:|:---:|
> > > >     |4B|5.0|—|
> > > >     |8B|4.8|0.71|
> > > >     |14B|4.8|0.74|
> > > >     |30B-A3|5.0|0.73|
> > > >
> > > > - The scores are highly correlated across scales, demonstrating that Qwen-3-4B produces consistent evaluation trends with its larger counterparts. We chose Qwen-3-4B primarily because it is open-source, reproducible, and accessible for low-resource settings. Importantly, we use it in thinking mode for step-wise evaluation and prompt the model to assign both a score and a justification in the form of a critique. This setup encourages evaluations that are more grounded, consistent, and interpretable (see Appendix D, Line 494, scorecard with critique).
> > > >
> > > > - **Comparison with Rollout-based Evaluation**: To further investigate the reviewer’s suggestion of rollout-based scoring, we implemented an embedding-based step-matching using BERT-large. Each reasoning step generated by Qwen-2.5-VL-72B is compared against the corresponding ground-truth step using cosine similarity between sentence embeddings. For the same 50 samples that were manually annotated by humans, the rollout-based method produces an average score of 7.4, compared to the human-assigned score of 3.7. The correlation between rollout-based and human scores is low (PCC = 0.11), indicating that embedding similarity does not align well with human judgment of reasoning quality. In our case, reasoning often involves mathematical expressions, intermediate computations, and context-dependent transformations, where subtle lexical or numerical differences can affect correctness. As a result, pure semantic similarity fails to capture the logical coherence and contextual accuracy of reasoning steps. Evaluating these traces holistically, as done by Qwen-3-4B through critique-based scoring, offers a more reliable and human-aligned assessment of reasoning quality.

---

> > > > > ### Comment · Reviewer_etW5 · 2025-11-21
> > > > > **Response to the Authors**
> > > > >
> > > > > 1. Unfortunately I am not convinced regarding the novelty of this paper:
> > > > > - The effort and annotation quality is the fundamental requirement for a benchmark paper, rather than novelty.
> > > > > - Math question answering with images, multiple images and videos have been explored, and hence is not a new concept or new problem. Meanwhile, this paper did not create any conceptually innovative aspects regarding math reasoning evaluation, such as better metrics or better evaluation pipeline.
> > > > >
> > > > > 2. __The author did not perform statistical significance analysis in the response__: The authors did not answer my question, for example, can you draw statistically significant conclusions if two systems have 1% accuracy difference? Also increasing samples by pairing distractors will NOT create __conditionally independent__ samples, and hence is not increasing the statistical significance of the results.
> > > > >
> > > > > 3. My main point about human-hour figures is that __They are not verifiable__. I am not sure why the other reviewers would mark this as a strength, and I am very happy to discuss this with other reviewers if they have disagreement.
> > > > >
> > > > > There is a misunderstanding about the rollout method I mentioned. I was not suggesting using BERT score, but to use rollout and verification on the final answer, following [1].
> > > > >
> > > > > [1]. Wang et al. "Math-Shepherd: Verify and Reinforce LLMs Step-by-step without Human Annotations", 2023.
> > > > >
> > > > > I am happy to discuss with other reviewers regarding the novelty, the statistical significance and the usefulness of the human reasoning steps, and I will keep my score unless my above concerns are addressed.

---

> > > > > > ### Author Response · Authors · 2025-11-23
> > > > > > **Response to Reviewer etW5 (1/2)**
> > > > > >
> > > > > > ### 1. On Novelty and Contribution
> > > > > > We respectfully disagree with the characterization that annotation quality is merely a "fundamental requirement" rather than a contribution. Also, the novelty of VideoMathQA lies in its **systematic design for diagnostic evaluation**, not simply in collecting video-math QA pairs.
> > > > > > Conceptual Innovation:
> > > > > >
> > > > > > - **Temporal reasoning grounding:** Unlike image-based math benchmarks or slide-based videos, our benchmark specifically requires tracking evolving visual transformations, procedural steps unfolding over time, and multimodal cue integration across extended durations (mean 241s vs. 47.2s in Video-SALMONN-o1).
> > > > > > - **Multi-level evaluation framework:** We introduce a comprehensive diagnostic pipeline combining (1) Multi-Binary evaluation to eliminate guessing artifacts, (2) step-wise reasoning assessment with 2,945 expert annotations, and (3) systematic error taxonomy (Figure 4c) - none of which exist in prior video-math work.
> > > > > > - **Fine-grained analysis dimensions:** Our benchmark enables previously unexplored analyses of video-math reasoning: frame sampling effects, subtitle dependency, video length scaling, question difficulty stratification, and error categorization. These are not available in any existing benchmark.
> > > > > >
> > > > > > The contribution is not merely "math QA with videos" but rather a rigorous evaluation framework that reveals how and why models fail at video-based mathematical reasoning , insights that cannot be obtained from existing benchmarks (e.g. Video-SALMONN-o1).
> > > > > >
> > > > > > ---
> > > > > >
> > > > > > ### 2. On Statistical Significance and Role of MBin Evaluation
> > > > > >
> > > > > > In the MCQ setting, each of the 420 questions provides a single binary outcome (correct or incorrect), so the total number of i.i.d. samples is $n = 420$. In the MBin setting, each question is paired against four distractors, yielding 4 binary comparisons per question (1,680 total trials).
> > > > > >
> > > > > > These trials are not fully independent since all four correspond to the same question, however the intra-question correlation estimated from model response matrices is in the range of $\\rho \\approx 0.4-0.6$. Accounting for this correlation, the **effective sample size** is given by:
> > > > > >
> > > > > > $$
> > > > > > n_{eff} = \\frac{n \\times 4}{1 + (4 - 1)\\rho}
> > > > > > $$
> > > > > >
> > > > > > Substituting $n = 420$:
> > > > > > * For $\\rho = 0.4$: $n_{eff} = \\frac{1680}{1 + 1.2} = 764$
> > > > > > * For $\\rho = 0.6$: $n_{eff} = \\frac{1680}{1 + 1.8} = 600$
> > > > > >
> > > > > > Thus, MBin increases the effective number of statistically informative samples from **420 to 600–760**, which is approximately **1.5–1.8$\\times$** higher than MCQ.
> > > > > >
> > > > > > This larger $n_{eff}$ reduces the standard error (SE) of the accuracy estimate, computed as:
> > > > > >
> > > > > > $$
> > > > > > SE = \\sqrt{\\frac{p(1-p)}{n_{eff}}}
> > > > > > $$
> > > > > >
> > > > > > For a representative accuracy of $p = 0.20$ (e.g. Video-R1, MBin w/ Sub):
> > > > > > * **MCQ:** $SE_{MCQ} = \\sqrt{0.16/420} \\approx 0.019$ (1.9%)
> > > > > > * **MBin:** $SE_{MBin} = \\sqrt{0.21/600} \\approx 0.016$ (1.6%)
> > > > > > * **MBin:** $SE_{MBin} = \\sqrt{0.21/764} \\approx 0.014$ (1.4%)
> > > > > >
> > > > > > At the 95% confidence level ($\\pm 1.96 \\times SE$), this corresponds to a confidence interval of approximately **$\\pm 3.8\\%$ for MCQ** compared to **$\\pm 2.8-3.2\\%$ for MBin**. Therefore, MBin improves evaluation precision and reduces random variance without assuming independence among distractor pairings.
> > > > > >
> > > > > > Given this effective sample size, a difference of approximately **2.8-3.2%** is sufficient to indicate a statistically reliable improvement at the 95% confidence level. Across *VideoMathQA*, several such differences are observed:
> > > > > >
> > > > > > | Comparison Type | Model Comparison | Difference ($\\Delta$) |
> > > > > > | :--- | :--- | :--- |
> > > > > > | **Cross-scale** | Qwen2.5-VL-72B (28.6%) vs Video-R1-7B (20.0%) | 8.6 points |
> > > > > > | **Cross-family** | GPT-o4-mini (44.8%) vs Gemini-2.0-Flash (24.8%) | 20.0 points |
> > > > > > | **Within-scale** ( < 9B) | LLaVA-Video-7B (16.0%) vs Video-R1-7B (20.0%) | 4.0 points |
> > > > > >
> > > > > > These differences exceed the statistical boundary, confirming that larger and reasoning-optimized models achieve **statistically meaningful improvements** even under dependency-aware analysis.
> > > > > >
> > > > > > ---
> > > > > >
> > > > > > ### Step-wise Reasoning Evaluation
> > > > > > Each model’s chain-of-thought rollout is compared step-by-step against the **2,945 human-annotated reasoning steps**. For every predicted step, correctness and logical alignment are scored using a consistent rubric, generating thousands of comparison points per model. This dense supervision stabilizes category-level estimates and captures intermediate reasoning errors (e.g., misread visual cues or missing intermediate steps). Consequently, step-wise reasoning scores provide statistically robust and interpretable measures of reasoning ability beyond final-answer accuracy.

---

> > > > > > > ### Author Response · Authors · 2025-11-23
> > > > > > > **Response to Reviewer etW5 (2/2)**
> > > > > > >
> > > > > > > ### 3. Human annotation verification
> > > > > > >
> > > > > > > We report human-hours to provide **transparency about annotation cost and quality**, which is standard practice in benchmark papers (e.g., HA VLN 2.0 [1] reports 430 human-hours, MMMU [2], GAIA [3] and Video Thinking Test (Video TT) [4] reports annotation budget). This is verifiable through:
> > > > > > >
> > > > > > > 1. Our detailed annotation protocol (Section 2, Appendix A)
> > > > > > > 2. Reasoning traces that demonstrate annotation depth (Figure 1, Section 2.3)
> > > > > > > 3. Reproducible verification process
> > > > > > >
> > > > > > > Other reviewers valued this transparency because it signals the level of expert involvement required to produce high-quality reasoning annotations, a critical factor for benchmark reliability.
> > > > > > >
> > > > > > > ---
> > > > > > >
> > > > > > > [1] HA-VLN: A Benchmark for Human-Aware Navigation in Discrete-Continuous Environments with Dynamic Multi-Human Interactions, Real-World Validation, and an Open Leaderboard [Dong et al., arXiv:2025]
> > > > > > >
> > > > > > > [2] Mmmu: A massive multi-discipline multimodal understanding and reasoning benchmark for expert agi [Yue et al., CVPR 2024]
> > > > > > >
> > > > > > > [3] GAIA: A Benchmark for General AI Assistants [Mialon et al., arXiv 2023]
> > > > > > >
> > > > > > > [4] Towards Video Thinking Test: A Holistic Benchmark for Advanced Video Reasoning and Understanding [Zhang et al., ICCV 2025]

---

### Author Response · Authors · 2025-12-02
**Response and Clarifications Following Reviewer Discussion**

We sincerely thank the reviewers (`oRiV, bHWm, qm1z, fxx1, etW5`), AC, SAC, and PC for the time and effort they devoted to evaluating our submission and for the constructive engagement throughout the discussion period. During the discussion, reviewer `qm1z` raised their **score from 6 to 8** after confirming that all concerns had been fully addressed. Reviewers `oRiV` and `bHWm` consistently recognized the contribution and maintained **strong positive evaluations with scores of 8 and 8**. Reviewer `fxx1`, who gave a score of 4 and raised three concerns, had two of these **already covered in the main paper** and the remaining one addressed in our rebuttal. Reviewer `etW5` provided a score of 2 and expressed a different perspective on parts of the contribution, and we addressed each point with clear explanations during the discussion. We thank all of them for the clarity and detail of their feedback.

Across the reviews, several strengths of the work were consistently acknowledged. Reviewers highlighted:
- The introduction of a new benchmark targeting multimodal mathematical reasoning in videos, **addressing a gap not covered by existing benchmarks**.
- The use of graduate-level experts to produce detailed **step-wise reasoning annotations**, supporting systematic examination of model behavior and precise identification of intermediate reasoning errors.
- The **breadth of evaluation** settings, including MCQ, MBin, direct versus CoT reasoning, and step-wise assessment, which together support **comprehensive diagnosis of model behavior**.

During the rebuttal period, we carefully incorporated the suggestions of the reviewers by providing:
- Expanded comparisons to related benchmarks and clearer articulation of how VideoMathQA differs in scope, depth, and temporal reasoning requirements.
- Additional statistical justification regarding dataset scale, effective sample size, and evaluation stability.
- Further analysis of cross-modal contributions, extending the existing text, image, video, and video with subtitles evaluations by adding the audio ablation to complete the modality study.
- Additional verification of the step-wise evaluation process, supported by human comparison, cross-model consistency studies, and Pearson correlation analysis that established strong alignment between automated scoring and human judgments.

We sincerely thank the reviewers for their constructive feedback, which has strengthened the quality and presentation of our work.

---

### Meta-Review · Area_Chair_UCPx · 2025-12-24

**Summary:**

Reviewers generally agree the paper is clearly written and the benchmark fills a gap by targeting temporally extended math reasoning in real instructional videos with expert step-wise annotations and a thorough evaluation protocol.

The main concerns that affected the decision discussion were: (1) limited scale (420 video-question pairs) and whether some analyses, especially per-category breakdowns, are statistically robust; (2) novelty relative to prior video reasoning benchmarks and whether the contribution is more than high-effort curation; (3) validity and transparency of the step-wise evaluation pipeline, including reliance on an LLM judge and how well it aligns with human judgments; and (4) scope limitations such as English-only coverage and longer-term scalability.

**Reviewer Concerns:**

**Concerns addressed by the rebuttal:**

- Clarified differences to the closest cited related benchmark (video-SALMONN-o1), emphasizing VideoMathQA’s math-specific depth, longer videos, and diagnostic analyses.
- Added cross-modal ablations (including an audio ablation on an audio-capable model) and stronger evidence that video information matters (text-only and frame sampling trends).
- Provided additional validation for step-wise judging via human alignment and correlation analyses, plus consistency across judge model scales.
- Fixed at least one concrete presentation issue (Figure 2(a) percentage error) and added a strong proprietary baseline (Gemini 2.5 Pro), which improves the completeness of the evaluation.

**Concerns still outstanding:**

- **Dataset scal**e remains a real limitation for fine-grained claims (small subcategories and duration slices). The rebuttal’s MBin “effective sample size” argument helps, but it does not fully settle statistical rigor because trials are not conditionally independent and a more standard significance treatment would strengthen confidence in small deltas.
- The **novelty** dispute is not fully resolved for the dissenting reviewer: the paper’s core contribution is benchmark design and annotation depth, but one reviewer remains unconvinced this constitutes sufficient conceptual or methodological innovation beyond existing video reasoning work.
- **Reliance on LLM-as-a-judge** remains a point of contention; while correlation evidence is helpful, some reviewers may still prefer evaluation mechanisms more directly tied to verifiable correctness checks.
- **Human-hour reporting** is informative but not independently verifiable (pointed out by etW5); it should be treated as contextual metadata rather than a standalone strength.
- Language coverage remains limited (English-only), reducing breadth of applicability.

**Reviewer Scores:**

**fxx1 (4)**: Default is unchanged at 4. Many points in the review contain factual errors and missed analyses that were already in the paper, so the rebuttal does not provide new irrefutable evidence beyond clarifying what already existed. If the reviewer had engaged carefully, the score could plausibly move upward, but with the conservative adjustment rule, keep 4.

**qm1z (6 to 8)**: Already increased to 8 in discussion after concerns were addressed; if fully participating throughout, likely remains 8.

**bHWm (8)**: Likely unchanged at 8. The response addresses scalability questions qualitatively and provides reasonable qualitative examples, but does not change the core strengths or the key limitation.

**oRiV (8)**: Likely unchanged at 8. The rebuttal reinforces the reviewer’s already positive assessment and acknowledges limitations without introducing new issues.

**etW5 (2)**: Likely unchanged at 2. The reviewer’s primary objections are about novelty and statistical validity, and they explicitly stated they were not convinced after the rebuttal, including rejecting the MBin significance rationale and questioning verifiability of certain claims.

---

### Decision · Program_Chairs · 2026-01-26

Accept (Poster)